# Coordination Polymers Based on Highly Emissive Ligands: Synthesis and Functional Properties

**DOI:** 10.3390/ma13122699

**Published:** 2020-06-13

**Authors:** Anastasia Kuznetsova, Vladislava Matveevskaya, Dmitry Pavlov, Andrei Yakunenkov, Andrei Potapov

**Affiliations:** 1Kizhner Research Center, National Research Tomsk Polytechnic University, 30 Lenin Ave., 634050 Tomsk, Russia; akuznetsova@tpu.ru (A.K.); vvm40@tpu.ru (V.M.); dipavlov@tpu.ru (D.P.); avy31@tpu.ru (A.Y.); 2Nikolaev Institute of Inorganic Chemistry, Siberian Branch of the Russian Academy of Sciences, 3 Lavrentiev Ave., 630090 Novosibirsk, Russia

**Keywords:** coordination polymers, metal-organic frameworks, luminescence, fluorescence, sensing, naphthalene diimide, 4,4′-stilbenedicarboxylic acid, emissive ligands, carbazole, thiazole

## Abstract

Coordination polymers are constructed from metal ions and bridging ligands, linking them into solid-state structures extending in one (1D), two (2D) or three dimensions (3D). Two- and three-dimensional coordination polymers with potential voids are often referred to as metal-organic frameworks (MOFs) or porous coordination polymers. Luminescence is an important property of coordination polymers, often playing a key role in their applications. Photophysical properties of the coordination polymers can be associated with intraligand, metal-centered, guest-centered, metal-to-ligand and ligand-to-metal electron transitions. In recent years, a rapid growth of publications devoted to luminescent or fluorescent coordination polymers can be observed. In this review the use of fluorescent ligands, namely, 4,4′-stilbenedicarboxylic acid, 1,3,4-oxadiazole, thiazole, 2,1,3-benzothiadiazole, terpyridine and carbazole derivatives, naphthalene diimides, 4,4′,4′′-nitrilotribenzoic acid, ruthenium(II) and iridium(III) complexes, boron-dipyrromethene (BODIPY) derivatives, porphyrins, for the construction of coordination polymers are surveyed. Applications of such coordination polymers based on their photophysical properties will be discussed. The review covers the literature published before April 2020.

## 1. Introduction

Coordination polymers are solid-state structures consisting of repeating coordination units extending in one, two or three dimensions [1]. The first preparation and application of coordination polymers probably dates back to early 18th century, when German chemists accidentally discovered the Prussian blue dye [2]. Crystallographic studies, the first of which was carried out in 1936 [3] revealed Prussian blue to be a 3D coordination polymer {[Fe^III^_4_Fe^II^_3_(CN)_18_]·11.0H_2_O}_n_, in which the alternating Fe^3+^ and Fe^2+^ ions are linked by bridging cyanide ions [4].

Coordination polymers in which metal ions are linked by organic ligands into structures with potential voids are often referred to as metal-organic frameworks (MOFs) or porous coordination polymers [5,6,7,8,9]. The topology of coordination polymers can be tuned almost at will by careful choice of metal ions and organic linkers and a nearly infinite variety of structures can be obtained [10,11,12]. Functional properties of the coordination polymers include capacity to store gases [13,14], separate gas [15,16,17] and liquid [18,19,20] mixtures, water purification [21,22,23], catalytic [24,25,26,27,28] and electrochemical [29,30,31,32] activities, biomedical applications [33,34,35,36,37,38].

Luminescence is an important property of coordination polymers, often playing a key role in their applications. Luminescence is a non-coherent radiation that occurs upon the excitation of atoms, ions or molecules. Luminescence arises when certain transitions (called spontaneous radiative transitions) of these species from the states with higher energy to the states with lower energy, including the ground state, take place. Depending on the excitation method, different types of luminescence are differentiated. Thus, photoluminescence occurs upon excitation by an optical radiation (usually in UV range), electroluminescence—when excited by an electrical field. The processes that accompany the luminescence are often visualized in Jablonski diagrams (Figure 1). Absorption of light occurs in a very short femtosecond timeframe and correspond to the excitation of the particle from the ground state (S_0_) to an excited state (S_1_, S_2_, …). It should be noted that each state has its own set of vibrational levels, which are populated upon excitation with different probabilities and when combined, form an absorption spectrum. After the absorption of a photon, the most probable process is called the internal conversion or vibrational relaxation. This process is longer that the excitation (picosecond timeframe) and is accompanied by a structural relaxation of the excited molecule. The excess energy is converted into heat and the relaxation is thus a non-radiative process. The molecule can exist in this excited state for nanosecond and longer and then returns to the ground state, emitting a photon in a process called fluorescence. Other events that can occur after the excitation include non-radiative relaxation upon collision of the excited molecule with other particles or intersystem crossing to the lowest excited triplet state (T_1_). Relaxation from the triplet state to the ground state with photon emission is called phosphorescence. Transition back to the S_1_ state is also possible, followed by a delayed fluorescence.

Coordination polymers are complex systems consisting of metal ions, one or more ligand types, inclusion of solvent molecules or other guests in voids is also possible. Emission of light by the coordination polymers can arise from various types of electron transitions—intraligand (ligand-centered), metal-centered, metal-to-ligand and ligand-to-metal charge transfer (MLCT and LMCT), Figure 1. Electron transitions in guest molecules encapsulated in the pores of the coordination polymers can also influence their photophysical properties.

The photophysical properties of the coordination polymers are used to create electroluminescent materials for LEDs [39,40,41], as contrast agents in biomedical imaging, theranostics and photodynamic therapy [42,43]. In recent years, more attention is given to nonlinear optical properties of the coordination polymers, including the second harmonic generation, multi-photon absorption, upconversion luminescence and lasing [44,45,46,47,48,49]. The most extensive area of the use of the luminescent properties of the coordination polymers is the development of sensors for various analytes - cations and anions in aqueous and non-aqueous solutions [50,51,52], gases (oxygen, nitric(II) oxide, carbon monoxide, ammonia, water vapor, etc.) [53,54,55,56], volatile organic compounds (aromatic hydrocarbons, aromatic nitro compounds, amines, etc.) [57,58,59,60], biologically important compounds (vitamins, pharmaceutical substances, toxins, DNA and RNA) [61,62,63]. The analytical signal in sensors of this type, as a rule, is associated either with a decrease in the luminescence intensity in the presence of an analyte (the “quenching” effect), or with its increase (the “turn-on” effect). The wide range of applications and the variety of building blocks of luminescent coordination polymers causes a rapid increase in the number of publications on this topic in the last 10–15 years. Thus, the first works devoted to the study of the photophysical properties of the coordination polymers appeared in 1997 [64], in recent years 400–500 publications devoted to this area were published annually, and to date, more than 4300 works have already been published, according to Scopus search results using the query “(luminescent OR fluorecscent) AND ((MOF OR metal-organic framework) OR coordination polymer)” (Figure 2).

Recently, several reviews on luminescent coordination polymers were published, but almost all of them were devoted to their sensory properties [65,66,67,68]. In addition, in most reviews, the emphasis was placed on coordination polymers based on lanthanides with metal-centered luminescence [69,70,71], and only one work of 2019 was devoted to a review of luminescent MOFs based on transition metals, but its area was also limited by the sensory properties of MOFs with respect to biologically relevant metal ions [51]. Within this review, data on the coordination polymers with ligand-centered luminescence and their functional properties of will be surveyed. The classification of the coordination polymers will be based on the types of ligands responsible for the appearance of luminescent properties.

## 2. Coordination Polymers Based on 4,4′-Stilbenedicarboxylic Acid

As of this day, 4,4′-stilbenedicarboxylic acid (H_2_sdc, Scheme 1), is an organic linker in the array of dicarboxylates widely used for the construction of coordination polymers. It is often encountered as a part of reticular syntheses due to its predictable geometry and availability. Its relatively large conjugated electron system, as well as a certain degree of flexibility make it interesting for the synthesis of luminescent MOFs [72,73]. “Rigidifying” of the ligand conformation in the resultant MOF often leads to the enhancement of stilbene-based luminescence, which allows the preparation of highly emissive and stable materials [74].

Bauer et al. [75] prepared two H_2_sdc-based MOFs [Zn_3_(sdc)_3_(dmf)_2_]_n_ and {[Zn_4_O(sdc)_3_(dmf)]·CHCl_3_}_n_ by varying synthetic conditions. It was discovered that crystal structure packing density influenced π-π interligand interactions, more dense structure demonstrating a red-shift and broadening of the emission band compared to a less densely packed coordination polymer (441 nm and 390 nm correspondingly). The emission of both MOFs was ascribed to intraligand excitations. The characteristic lifetimes were longer compared to the free ligand, indicative of an increased rigidity of sdc^2-^ linkers in the coordination network. MOF formulated as [Zn_4_O(sdc)_3_(dmf)]·CHCl_3_}_n_ demonstrated luminescence sensitivity to inclusion of guest solvent molecules [75].

Several years later [76] the same group, supporting their earlier hypothesis, reported the synthesis of eight isostructural MOFs having the formula [(M1)(M2)_2_(sdc)_3_(dmf)_2_]_n_ (M1 = M2 = Zn^2+^ (1), Cd^2+^ (2), Mn^2+^ (3), Co^2+^ (4); M1 = Zn^2+^, M2 = Cd^2+^ (5), Mn^2+^ (6), Co^2+^ (7); M1 = Co^2+^, M2 = Mn^2+^ (8)). Compounds 1–3 and 5–7 had similar emission maxima near 440 nm, however, the emission intensities for MOFs 3, 6 and 7 were much lower. These differences were attributed to the occurrence of luminescence quenching in 3, 6 and 7 by through-space electron- and/or energy transfer, due to the proximity of the ligand to high-spin ions (Mn^2+^, S = 5/2 or Co^2+^, S = 3/2).

Liang et al. [77] prepared MOF [Zn(1,3-bpeb)(sdc)]_n_ (1,3-bpeb—1,3-bis[2-(4-pyridyl)ethenyl]benzene, Scheme 1) which demonstrated strong blue emission, however, no detailed study of photophysical properties was conducted.

Quah et al. [78] prepared three isostructural MOFs {[Zn_2_(sdc)_2_(An2Py)]·DMF·4H_2_O}_n_, {[Zn_2_(sdc)_2_(An2Py)]·perylene}_n_ and {[Zn_2_(sdc)_2_(An2Py)]·anthracene}_n_ (An2Py—*trans*,*trans*-9,10-bis(4-pyridylethenyl)anthracene, Scheme 1, Figure 3). Encapsulation of highly emissive organic chromophores with emission bands overlapping with the band of the host structure allowed preparation of the four-photon upconverting MOFs. Authors believe that enhancement of the multiphoton-excited photoluminescence in MOFs in comparison to organic ligands happens due to the rigidifying effect of the MOF, as well as Förster resonance energy transfer (FRET) between the host MOF and the guest molecules. Quantum yields of MOFs with encapsulated chromophores, however, were found to be on the lower end—25% and 26%.

Later the same authors have extended this study by synthesizing an additional series of MOFs [79]. Among them, the new structure with H_2_sdc as a ligand, {[Zn_2_(sdc)_2_(AnEPy)]·2DMA·1.5H_2_O}_n_ (AnEPy—*trans*,*trans*-9,10-bis(4-pyridylethynyl)anthracene) demonstrated an emission maximum at 560 nm and a quantum yield of 21%. The authors explored the effect of structural variation of MOFs on two-photon excited emission. However, it was found that both the quantum yields and two-photon absorption cross-sections did not exhibit comprehensible structure−property relationship.

Deng [80] prepared two highly interpenetrated structures with dia topologies, using Ni^2+^ and Zn^2+^ as metal centers. The emission peaks appeared at 398 nm (λ_ex_ = 328 nm) for MOF {[Zn(sdc)(bmp)]·H_2_O}_n_ (bmp—1,4-bis(2-methylimidazol-1-yl)benzene, Scheme 1) and at 393 nm (λ_ex_ = 303 nm) for MOF {[Ni(sdc)(bimb)]·DMF}_n_ (bimb–1,4-bis(imidazol-1-yl)-2,5-dimethylbenzene, Scheme 1), they were assigned to π-π* intraligand transitions. The author noted that π*-n carboxylate transitions in sdc^2−^ did not significantly contribute to the luminescence of MOFs in the presence of bmp N-donor ligand.

Self-catenated rob-type net {[Zn_2_(dmtrz)_2_(sdc)]·6H_2_O}_n_ (MAC-11, Hdmtrz—3,5-dimethyl-1H-1,2,4-triazole), consisting of Zn-triazolate 2-D layers linked together by H_2_sdc was described in [81]. It was found that the framework undergoes thermo-induced phase transformation, accompanied by a photoluminescence response. The authors assume that the observed red-shift of 35 nm (418 nm to 453 nm) is due to the flattening of Zn(dmtrz) layers, as well as the changes in the coordination mode of H_2_sdc, leading to the enhanced interaction between the carboxylate ligand and Zn^2+^ centers.

Further proof for the importance of the MOF rigidifying effect was given in the work of Fan et al., who prepared two interpenetrated Zn/Cd MOFs based on H_2_sdc and bmib (1,4-bis(2-methylimidazol-1-yl)butane, Scheme 1) or tib (1,3,5-tris(1-imidazolyl)benzene, Scheme 1) ligands [82]. Upon excitation at 250 nm, both MOFs exhibited strong emission (410 nm for Zn and 399 nm for Cd framework), which is close to the ligand emission maximum (381 nm).

Later the same authors studied two additional structures, namely {[Zn(sdc)(bmib)]·0.4H_2_O}_n_ and {[Zn(sdc)(bimb)]·DMF}_n_ (bimb—1,4-bis(imidazol-1-yl)butane, Scheme 1) [83]. Both structures displayed similar fluorescence emission band maxima at 384 nm (λ_ex_ = 245 nm) and 401 nm (λ_ex_ = 240 nm). The authors attribute the evident luminescence red-shift to the change of flexible bis(imidazole) ligand conformation.

One of the complexes prepared in [84] utilizes H_2_sdc as co-ligand for building of the Zn^2+^ based framework. Structure formulated as {[Zn_2_(sdc)_2_(bips)_2_]·5H_2_O}_n_ (bips—bis(4-(imidazol-1-yl)phenyl)sulfone, Scheme 1) exhibits a broad band with a peak at 446 nm upon excitation at 337 nm. Despite the presence of the luminescent dicarboxylate ligand, the authors ascribe the luminescence of the MOF to the bis(imidazole) ligand.

Barsukova et al. prepared a series of MOFs built from H_2_sdc and flexible bis(imidazol-1-yl)alkanes [85]. One of the coordination polymers, {[Zn(sdc)(bim)]·DMF}_n_, demonstrated one of the best values for MOF luminescence quantum yields—82% before activation (λ_em_ = 455 nm, λ_ex_ = 390 nm). Upon further investigation, the increase of the quantum yield of the sdc-based luminescence was attributed to the rigidifying of the framework with increased interpenetration.

One of the structures obtained in [86], {[Zn_4_(tppa)_2_(sdc)_3_(NO_3_)_2_]·4DMF·2MeCN}_n_ (tppa–tris(4-pyridylphenyl)amine), contains sdc^2−^ linkers, however, the authors attribute MOF luminescence (λ_em_ = 525 nm, λ_ex_ = 365 nm, quantum yield (QY) 21%) only to the N-donor ligand.

Several Zn-MOFs are described in [87]. Upon excitation at 350 nm, structures formulated as [Zn_3_(sdc)_3_(py)_2_]_n_, [Zn_3_(sdc)_3_(4,4′-bpy)]_n_, {[Zn_3_(sdc)_3_(bpea)]·3H_2_O}_n_ (py—pyridine, 4,4′-bipy—4,4′-bipyridine, bpea—1,2-bis(4-pyridyl)ethane, Scheme 1) exhibited emission bands at 460, 487 and 469 nm (Table 1), showing different degrees of blue shift compared to the free ligand. Authors state that the shift is due to the interchromophore coupling and metal coordination as well as intraligand π-π* transitions.

The authors of [88] studied the luminescent properties of two coordination polymers [Zn(sdc)(H_2_O)]_n_, [Cd(sdc)(H_2_O)]_n_, however, only data for the Zn structure is present. It demonstrated strong emission bands at 435 and 459 nm in the solid state at room temperature upon excitation at 387 nm. Increased emission intensity of the coordination polymers compared to the free ligand was attributed to a change of π*-π transitions of the free ligand to π*-n transitions upon coordination. Interestingly enough, the authors mention that the emission of Cd based coordination polymer is weak compared to the Zn one.

The structure formulated as [Zn(sdc)(bita)]_n_ (bita—4,4′-bis(imidazole-1-yl)triphenylamine, Scheme 1) demonstrated strong luminescence with a broad peak at 470 nm upon excitation at 367 nm. Ligand to ligand charge transfer is stated as the primary mechanism [89].

As one of goals of the work [90], luminescence properties of two polymers, [Zn_2_(sdc)_2_(pvsp)]_n_ (pvsp–4-(4-((E)-2-(pyridin-4-yl)vinyl)-styryl)pyridine) and {[Zn(sdc)(4,4′-bpy)]·2DMF}_n_, were evaluated. It was found that the first compound exhibits photoluminescence with the maximum emission at 484 nm (excited upon 425 nm) and the second at 505 nm (excited upon 417 nm). Mechanism of photoluminescence for both coordination polymers was assigned to ligand-to-metal charge transfer (LMCT).

In the course of preparation of entangled MOFs, Zn based MOF {[Zn_4_(sdc)_4_(beips)_2_]∙14DMF}_n_ (beips—bis(4-(2-ethylimidazol-1-yl)phenyl)sulfone) was characterized [91]. It demonstrated a two-band emission at 437 nm and 464 nm upon excitation at 406 nm.

Huang et al. reported an open framework [Cd(sdc)(H_2_O)]_n_ which exhibited strong luminescence at 460 nm upon excitation at 392 nm [92]. Enhancement of the luminescence in comparison to the free ligand was attributed to the rigidifying effect of the coordination network.

H_2_sdc was used as a linker in the synthesis of capsule-based MOF {[Cd(ttr4a)(sdc)]·1.5H_2_O}_n_ (ttr4a—tetrakis(1,2,4-triazol-1-ylmethyl)resorcin[4]arene) [93]. The resulting structure produced a strong emission band at 447 nm upon excitation at 367 nm. The emission enhancement was ascribed to the cooperative effect of the auxiliary ligand and H_2_sdc.

A pillar-layered coordination network {[Cd_3_(sdc)_2_(trz)_2_(H_2_O)_2_]·DMF}_n_ (Htrz—1,2,4-triazole) prepared by Xiao et al. exhibited strong emission band near 443 nm upon excitation at 320 nm, which was attributed to ligand-based fluorescence [94]. The excited state lifetimes of the coordination polymer in the solid-state were 1.05 ns (88%) and 2.29 ns (12%). The quantum yield of the solid-state luminescence was determined to be 38%. This compound has a potential application in sensor systems, since it was found that exposure to toluene significantly enhances the luminescence intensity, while exposure to nitrobenzene quenched it to a significant extent. The authors note that the outstanding sensing capability of this complex may be attributed to its infinite 3D framework structure and the photoinduced charge transfer electron transitions between MOF and solvent molecules.

Wang et al. reported three Cd MOFs [Cd_3_(bdc)_3_(dma)_4_]_n_, {[Cd_2_(tdc)_2_(dma)_2_]·DMA}_n_ and [Cd_3_(sdc)_3_(dma)]_n_ (H_2_bdc—terephthalic acid, H_2_tdc—thiophenedicarboxylic acid) [95], one of which contained sdc^2−^ linkers. The photoluminescent properties were surveyed, the coordination polymers demonstrated emission maxima attributed to inter- or intra-ligand-based electron transitions at 450, 470 and 455 nm, correspondingly.

In the work related to the preparation of interlocked architectures luminescence properties of the coordination polymer {[Cd_3_(sdc)(phen)_3_(OH)_3_(H_2_O)]·0.5 sdc·4H_2_O}_n_ (phen—1,10-phenanthroline) were studied [96]. It exhibited an intense blue radiation with emission maximum at 489 nm upon excitation at 386 nm. The excited state lifetime of about 14 ns was significantly longer compared to the systems without hydroxo-metal clusters, which was explained by the presence of μ_3_-OH ligands in the structure.

Golafale et al. synthesized two isostructural lanthanide based and one Cd based frameworks with 2,2′-diamino-4,4′-stilbene dicarboxylic acid (H_2_sdcNH_2_) as a linker (Figure 4) [97]. The coordination polymers {[Ln(sdcNH_2_)(HCOO)(H_2_O)]·2DEF}_n_ (Ln = Yb^3+^ and Tm^3+^) and [Cd_2_(sdcNH_2_)(NO_3_)_2_(dmf)_4_]_n_ displayed strong luminescence with emission maxima matching those reported for diaminostilbenes. Yb framework exhibited an emission maximum at 528 nm (excitation 382 nm, Stokes shift 146 nm), while for Tm MOF the maximum was at 518 nm (excitation 345 nm, Stokes shift 173 nm). While Stokes shifts for Cd framework were somewhat lower (emission maxima at 475 and 495 nm, excitation 351 and 376 nm, Stokes shift 119 and 124 nm), it demonstrated the highest intensity and radiative lifetime (triexponential function, τ_1_ = 0.80 ns, τ_2_ = 0.13 ns and τ_3_ = 2.48 ns). The authors attributed the differences in photophysical properties to the structural peculiarities—the amino group in Cd framework is coordinated, further rigidifying the ligand in comparison to Ln MOFs and reducing the non-radiative energy loss.

The same group reported two lanthanide-based MOFs, [Er_3_O_5_(sdc)_4_]_n_ and [Tm_3_O_5_(sdc)_4_]_n_, data for the photoluminescence of the Tm^3+^ coordination polymer were presented [98]. Upon excitation at 341 nm, it exhibits a broad band with a maximum at 475 nm. Stokes shift was smaller compared to the H_2_sdc powder, which is consistent with the data for Zn-sdc MOFs. Radioluminescence was also surveyed, and it was found that there were little differences between photo- and radioluminescence spectral profiles, which is indicative of minimal to no structural changes upon exposure to ionizing radiation. In addition, time-resolved experiments were carried out.

Using a thioether-decorated H_2_sdc (H_2_MeS-sdc, Scheme 1) as well as a pristine one, Li et al. engineered four lanthanide coordination polymers {[Tb_2_(sdc)_3_(dmso)_4_]·DMSO}_n_, {[Eu_2_(sdc)_3_(dmso)_4_]·DMSO}_n_, [Eu_2_(MeS-sdc)_3_(dmf)_2_(H_2_O)_2_]_n_ and [Tb_2_(MeS-sdc)_3_(dmf)_2_(H_2_O)_2_]_n_ [99]. Photoluminescent properties of all compounds were explored, and it was found that MOFs with the pristine H_2_sdc exhibit broad ligand-based emission bands with a maximum near 447 nm. On the contrary, emission of the coordination polymers constructed from the decorated H_2_sdc and Eu^3+^ ions comprised both the ligand-based and the metal-based luminescence (Table 1). The sharp lines at 393, 464 and 535 nm in the excitation spectrum were assigned to the metal-centered transitions ^7^F_0,1_→^5^L_6_, ^7^F_0,1_→^5^D_2_, ^7^F_0,1_→^5^D_1_, respectively, while the ligand was responsible for the weak broad band in at 370 nm. The sharp emission peaks at 579, 591, 615 and 696 nm in the emission spectrum were determined to be characteristic for Eu-centered transitions ^5^D_0_→^7^F_0_
^5^D_0_→^7^F_1_, ^5^D_0_→^7^F_2_, and ^5^D_0_–^7^F_4_ respectively. Recording the second emission spectrum at 350 nm excitation wavelength, the authors confirmed the occurrence of LMET with low efficiency. Thioether-decorated MOF [Tb_2_(MeS-sdc)_3_(DMF)_2_(H_2_O)_2_]n showed broad ligand-based excitation in the range of 350–450 nm and emission at 475–600 nm, while the direct f–f transition of Tb^3+^ occurred upon 487 nm excitation (^7^F_6_→^5^D_4_) and the emission peaks at 542 nm, 584 nm, 620 nm on emission were assigned to ^5^D_4_→^7^F_5_, ^5^D_4_→^7^F_4_, ^5^D_4_→^7^F_3_ transitions, respectively. The authors conducted preliminary luminescence response tests to Hg^2+^ and Cd^2+^ ions in solution, but the coordination polymers remained inert to the presence of these ions under the conditions studied [99].

Huang et al. studied the reversible phase transitions in two manganese based coordination polymers [100]. By adjusting the temperature and water content in the solvent mixture it was possible to selectively isolate [Mn_3_(sdc)_3_(H_2_O)_2_]_n_ and [Mn(sdc)(H_2_O)_2_]_n_. Both compounds exhibited good luminescence upon excitation at 347 nm with the emission maxima at 447 and 466 nm, respectively. The water-stable MOF [Mn(sdc)(H_2_O)_2_]_n_ was employed as a luminescent sensor for Pb^2+^ ions in water, with the detection limit of 31.4 nM. The luminescence quenching was explained by binding between Pb^2+^ ions and the framework, confirmed by XPS [100].

## 3. Coordination Polymers Based on 1,3,4-Oxadiazole Derivatives

1,3,4-Oxadiazoles are well-known for their range of biological activities [101], but also they along with their coordination compounds are extensively used in highly emissive materials [102,103].

Guo et al. synthesized luminescent MOF {[Zn_2_(mfda)_2_(4-bpo)(H_2_O)]·DMF}_n_ (H_2_mdfa—9,9- dimethylfluorene-2,7-dicarboxylate anion, 4-bpo—2,5-bis(4-pyridyl)-1,3,4-oxadiazole, Scheme 2), exhibiting parallel mutual polythreadings of 2D layers which are connected by hydrogen bonds into a self-penetrating framework with (4^4^∙6^10^∙7)(4∙5∙6)(4^6^∙5^2^∙6^16^∙7^1^∙9) topology [104]. Upon excitation at 358 nm, MOF demonstrated a strong blue emission with a broad band at 456 nm in solid-state photoluminescence spectrum. The free 4-bpo ligand showed emission near 385 nm in the solid state. Red-shift of emission of almost 100 nm was attributed to the ligand-to-metal charge transfer (LMCT).

Du et al. prepared Co^II^, Cu^II^, Zn^II^, Cd^II^ coordination assemblies based on 4- and 3-bpo (2,5-bis(3-pyridyl)-1,3,4-oxadiazole) [105]. Only [Cd(4-bpo)(dca)_2_]_n_ (dca–dicyanamide anion, N(CN)_2_^−^) had a 3D coordination motif with self-penetrating architecture. Solid-state PL spectrum of this coordination polymer demonstrated emission at 522 nm with a significant LMCT-induced red-shift (105 nm) relative to the free 4-bpo. Coordination polymers based on 3-bpo showed emission peaks at 361 nm for Zn^II^ and 365 nm for Cd^II^ compounds upon excitation at 345 nm.

Li et al. prepared a number of coordination polymers based on Co^II^, Ni^II^, Zn^II^, Ag^II^, Cd^II^, Cu^II^ and Pb^II^, tetrabromoterephthalic acid (H_2_tbta) and 4-bpo as a co-ligand [106]. All of these coordination polymers had various coordination motifs and dimensionality (1D to 3D). Compound [Pb(tbta)(4-bpo)]_n_ comprised a 3D coordination network with **dia** topology and exhibited fluorescent emission at 502 nm with excitation maximum at 342 nm. Similarity between the emission profiles of the ligand and MOF allowed to tentatively ascribe the photoluminescence mechanism to intraligand excitation.

Fang et al. reported two new copper(I) coordination polymers [Cu_3_(4-bpo)I_3_]_n_ and [Cu_3_(4-bpt)I_2_]_n_ (4-bpt—3,5-bis(4-pyridyl)-1,2,4-triazolate) [107]. It is interesting to note that 4-bpt based MOF formed as a result of in situ transformation of 4-bpo to 4-bpt in aqueous ammonia (Figure 5). Coordination polymer [Cu_3_(4-bpo)I_3_]_n_ had a 2D topology, while the 1,2,4-triazolate compound [Cu_3_(4-bpt)I_2_]_n_ was a 3D MOF. [Cu_3_(4-bpo)I_3_]_n_ displayed a strong red luminescence at 648 nm upon excitation at 465 nm, presenting the first example of red luminescent coordination compound based on 1,3,4-oxadiazole derivative and CuI. The coordination polymer demonstrated multicomponent fluorescent lifetimes of τ_1_ = 0.05, τ_2_ = 0.13 and τ_3_ = 0.01 µs, quantum yields 46.74%, 42.59% and 10.48%, respectively.

Chen et al. prepared Cd^II^ and Zn^II^ coordination polymers {[Cd(3-bpo)(mip)(H_2_O)]·2H_2_O}_n_, {[Cd(4-bpo)(hip)(H_2_O)]·4H_2_O}_n_ and {[Zn(4-bpo)(bdc)]·CH_3_OH}_n_ (H_2_mip—5-methylisophthalic acid, H_2_hip—5-hydroxylisophthalic acid, H_2_bdc—terephthalic acid) [108]. Excitation of the Cd^II^ coordination polymers at 350 nm leads to a fluorescent emission with peaks at 425 nm (for 4-bpo) and 378 nm (for 3-bpo) nm. It is interesting to note that no emission was observed for {[Zn(4-bpo)(bdc)]·CH_3_OH}_n_. The authors attribute the quenching effect to high-energy C-H and/or O-H oscillators in MeOH lattice molecules.

## 4. Coordination Polymers Based on Sulfur Heterocyclic Derivatives

Luminescent materials are often based on sulfur heterocycles, among which 4-hydroxythiazole is worth noting as responsible for the remarkable bioluminescence phenomenon [109].

### 4.1. Thiazole Derivatives

Zhai et al. prepared a series of Zn^II^/Cd^II^ MOFs based on 2,5-bis(4-pyridyl)thiazolo[5,4-d]thiazole (Py_2_TTz, Scheme 3) and various dicarboxylic acids—{[Zn_2_(Py_2_TTz)(2-CH_3_-bdc)_2_]∙2DMF∙3H_2_O}_n_, {[Cd_2_(Py_2_TTz)_2_(2-CH_3_-bdc)_2_]·2DMF·2EtOH}_n_, {[Zn(Py_2_TTz)_0.5_(2,6-ndc)]·DMF·EtOH}_n_, {[Zn(Py_2_TTz)(1,4-ndc)]·DMF·H_2_O}_n_, {[Zn_0.5_(Py_2_TTz)_0.5_(2,5-di-CH_3_-bdc)_0.5_]·DMF}_n_, {[Cd_2_(Py_2_TTz)_2_(2,6-ndc)_2_]∙3DMF∙4H_2_O}_n_ (2-CH_3_-H2bdc—2-methyl-1,4-benzenedicarboxylic acid, 2,5-di-CH_3_-H_2_bdc—2,5-dimethyl-1,4-benzenedicarboxylic acid, 1,4-H_2_ndc—1,4-naphthalenedicarboxylic acid, 2,6-H_2_ndc—2,6-naphthalenedicarboxylic acid, Scheme 3) [110]. All of these MOFs displayed good emission in the solid-state at room temperature from ≈460 nm to ≈560 nm (Table 2). The free ligand Py_2_TTz showed two emission peaks at 439 nm and 452 nm upon excitation at 409 nm. The observed red-shift of MOF emission was attributed to the LMCT.

Han et al. prepared Zn^II^, Ni^II^ and Cd^II^ MOFs based on V-shaped thienylpyridyl ligand (bptp, Scheme 3) and their application for sensing of metal ions was investigated [111]. 4,4′-Oxyidibenzoic acid (H_2_oba, Scheme 3) was used as co-ligand. The solid-state PL spectrum of bptp showed an emission at 474 nm upon excitation at 370 nm, while Zn-MOF [Zn_2_(oba)_2_(bptp)]_n_ showed a blue-shift of 43 nm upon excitation at 396 nm, Ni-MOF [Ni(oba)_2_(bptp)_2_(H_2_O)_2_]_n_ demonstrated a 13 nm red-shift upon excitation at 408 nm. Cd-MOF [Cd_2_(oba)_2_(bptp)(H_2_O)]_n_ displayed a red-shift of 15 nm upon excitation at 391 nm and was the most effective detection of Fe^3+^ and Al^3+^ ions in 5 × 10^−3^ M solutions in DMF. It is interesting to note that Fe^3+^ ions had a dramatic luminescence quenching effect, while Al^3+^ caused luminescence intensity enhancement [111]. XPS measurements showed weak interaction between Fe^3+^ and nitrogen atoms of free pyridyl groups, which was proposed as responsible for luminescence quenching.

### 4.2. Derivatives of 2,1,3-Benzothiadiazole

A series of luminescent MOFs based on 2,1,3-benzothiadiazole derivatives is reported in [112]. In this work, Cd^II^, Zn^II^, Co^II^, Ni^II^ coordination polymers were synthesized under solvothermal conditions with 4,7-bis(4-pyridyl)-2,1,3-benzothiadiazole (dpbt, Scheme 4) as a ligand. Terephthalic (H_2_bdc) and isophthalic acids (H_2_ipa) were used as co-ligands. Solid-state PL spectrum of the free dpbt shows emission at 465 nm upon excitation at 371 nm. {[Cd(dpbt)(bdc)]·2H_2_O}_n_ shows emission at 464 nm upon excitation at 356 nm and {[Zn_2_(dpbt)_2_(ipa)_2_]·2DMA}_n_ displays emission at 513 nm upon excitation at 375 nm. The emission of Cd-MOF was assigned to n*-π and/or π*-π transitions, while the emission red-shift of Zn-MOF allowed to assume LMCT contribution. Zn^II^-MOF was demonstrated to be perspective for detection of nitro compounds. The fluorimetric titration of MOF suspension by the addition of nitro compounds results in quenching of the MOF photoluminescence. The highest PL quenching degree of as high as 99.7% was observed for the picric acid at the concentration of 0.1 mM. Strong quenching was explained a lower LUMO energy level of PA in comparison to that of dpbt, while other nitro compounds demonstrated higher LUMO levels. Similar results were obtained for Cd^II^-MOF [112].

Coordination polymers {[Zn_4_(dpbt)_2_(bdc)_4_]∙H_2_O∙2dpbt)}_n_, {[Cd_2_(dpbt)_2_(bdc)_2_]∙DMA}_n_ demonstrated high selectivity for picric acid detection, as well as detection of metal ions [113]. The emission mechanism was assigned to ligand-centered with LMCT contribution. Zn-MOF demonstrated significant PL quenching in the presence of Fe^3+^ ions and PL intensity enhancement in presence of Al^3+^ and Cr^3+^ ions. Increasing the concentration of Fe^3+^ ions up to 0.9 mM lead to complete quenching of Zn-MOF emission. The emission intensity progressively increased with the concentration increase from 0.01 mM to 0.5 mM for Al^3+^ ions and 0.01 mM to 0.18 mM for Cr^3+^ ions. Cd-MOF displayed PL enhancement in presence of Al^3+^, Cr^3+^ and Pb^2+^. For this MOF the concentration range for Al^3+^ was 0.01–0.15 mM, 0.03–0.15 mM for Cr^3+^ and 0.01–0.1 mM for Pb^2+^ ions [113]. The emission enhancement in the presence of low concentrations of Al^3+^ and Cr^3+^ ions was tentatively ascribed to their coordination to dpbt ligands.

Song et al. prepared four zinc coordination polymers based on 4,7-bis((E)-2-(pyridine-4-yl)vinyl)-2,1,3-benzothiadiazole (bptda, Scheme 4) with diverse topologies [114]. Polycarboxylate possessing different bend angles (4,4′-dicarboxydiphenylamine (H_2_dpa), 4,4′-oxybisbenzoic acid (H_2_oba) and 4,4′-sulfonyldibenzoic acid (H_2_sdba)) were used as co-ligands, coordination polymers [Zn(bptda)(dpa)_2_]_n_, {[Zn_2_(bptda)(oba)_2_]∙2.75DMF}_n_, {[Zn_4_(bptda)_3_(oba)_4_]∙2H_2_O}_n_ and [Zn_2_(bptda)(sdba)_2_]_n_ were obtained. The PL spectra of these MOFs showed emission and excitation bands close to those of the free bptda ligand with emission at 560 nm and excitation at 367 nm. The photocatalytic degradation of rhodamine B (RhB), a common dye pollutant, in aqueous solution in the presence of all MOFs was explored, the highest degradation degree of 77% after 180 min irradiation (30 W LED lamp) was observed for {[Zn_2_(bptda)(oba)_2_]∙2.75DMF}_n_.

Cheng et al. prepared two Cd^II^ MOFs based on 2,1,3-benzothiadiazole-4,7-dicarboxylic acid (H_2_btdc, Scheme 4), {[S@Cd_6_(btdc)_6_]∙9H_2_O}_n_ and {[S@Cd_6_(btdc)_6_]∙6H_2_O}_n_ [115]. The free H_2_btdc ligand shows emission near 470 nm upon excitation at 260–440 nm in a mixed solvent (DMF/EtOH/H_2_O, 2.5:2:0.5). Both MOFs show emission at ~450 nm upon excitation at 370 nm in the solid-state. PL solvatochromism was demonstrated for MOF dispersions. Thus, in dioxane, petroleum ether, toluene, EtOH emission blue-shifts relative to the solid-state PL were observed, while in ethylene glycol, ethyl acetate and MeOH red-shifts were detected. In acetone the emission spectrum presented a two-shoulder band.

A benzothiadiazole-decorated UiO-68 was synthesized by Mallick et al. [116]. In this work luminescent analogue of *p*-terphenyl-4,4″-dicarboxylic acid, 4,4′-(2,1,3-benzothiadiazole-4,7-diyl)dibenzoic acid (H_2_btdb, Scheme 4) was used as a ligand (Figure 6). The free ligand shows emission at 480 nm upon excitation at 365 nm in water, while a water suspension of MOF Zr-btdb-**fcu** demonstrates a 21 nm red-shift and emission at 501 nm. Additionally, studies of the luminescent sensing of volatile organic amines (methylamine, ethylamine, triethylamine, aniline, *p-*phenylenediamine) in aqueous solution were performed. Increasing the concentrations of aniline and *p*-phenylenediamine led to a drastic decrease of MOF fluorescence intensity. In contrast, fluorescence turn-on effect was observed in the presence of aliphatic amines such as methylamine [116]. The turn-on effect was explained by the bonding between the protonated methylamine molecules and nitrogen atoms of btdb^2−^ linkers, leading to suppressed twisting motion of the ligand, which reduces the probability of nonradiative relaxation pathways. Interaction between the protonated methylamine and the framework was confirmed by DFT calculations.

The same ligand H_2_btdb in combination with 4,4’-(1H-benzo[d]imidazole-4,7-diyl)dibenzoic acid was used for the preparation of mixed-ligand Zr-**fcu**-MOF in which energy transfer between the linkers was observed [117]. A strong overlap between the emission band of benzimidazole ligand and the absorption band of H_2_btdb ensured a superior efficiency of energy transfer of 90%, making it a promising light-harvesting platform.

Zn-MOF with H_2_btdb ligand was prepared by Wei et al. [118]. This MOF exhibited strong luminescent properties both in the solid state and in MeOH suspension. In the solid-state {[Zn(btdb)(DMA)]∙H_2_O}_n_ exhibited an intense emission band at ~494 nm upon excitation at 370 nm, in methanol suspension it demonstrated emission at 491 nm upon excitation at 350 nm. As it was shown in the article, Zn-MOF had luminescent response for Cd^2+^ ions in methanol. The emission intensity enhanced distinctly in the presence of Cd^2+^ ions, thus, upon addition of 3 equivalents of Cd^2+^ the emission intensity increased by 3.5 times [118].

Tian et al. prepared Co^II^ MOF based on 4,7-bis(1*H*-imidazol-1-yl)-2,1,3-benzothiadiazole (bibt, Scheme 4) and 1,3,5-benzenetricarboxylic acid (H_3_btc) as co-ligands [119]. MOF {[Co_3_(bibt)_3_(btc)_2_(H_2_O)_2_]·solvents}_n_, JXUST-2 exhibited emission at 396 nm in solid state. After being immersed in DMA it showed emission peak at 530 nm upon excitation at 394 nm. The free bibt ligand demonstrated emission band around 540 nm upon excitation at 394 nm. JXUST-2 was tested for selective detection of metal ions and exhibited a turn-on response to Fe^3+^, Cr^3+^ and Al^3+^.

## 5. Coordination Polymers Based on Nitrogen Heterocycles

### 5.1. Terpyridine Derivatives

An et al. prepared cadmium(II)-terpyridine coordination frameworks with different carboxylate ligands [120]. The free 4′-(4-pyridyl)-4,2′:6′,4′′-terpyridine (4-pytpy, Scheme 5) ligand displayed emission at ~380 nm upon excitation at 310 nm (Table 3). For the MOFs there were emission bands at 523 nm for {[Cd(4-pytpy)(1,4-ndc)]∙1.5H_2_O}_n_, 526 nm for {[Cd(4-pytpy)(2,5-tdc)]∙H_2_O}_n_ and 373 nm for [Cd_2_(4-pytpy)(sdba)_2_]_n_ (1,4-H_2_ndc—1,4-naphthalenedicarboxylic acid, 2,5-H_2_tdc—2,5-thiophenedicarboxylic acid, H_2_sdba—4,4′-sulfonyldicarboxylic acid) under excitation at 310 nm. A blue-shift observed for the last MOF was assigned to the intraligand transfer (n-π* or π-π*). A red-shift of emission of other studied MOFs was ascribed to LMCT mechanism.

Song et al. prepared two Zn^II^-MOFs based on isomeric terpyridines, {[Zn(2-pytpy)(fum)]∙H_2_O}_n_ and {[Zn_6_(4-pytpy)_3_(mal)_4_]∙5H_2_O}_n_ (2-pytpy—4′-(4-pyridyl)-2,2′:6′,2″-terpyridine, 4-pytpy—4′-(4-pyridyl)-4,2′:6′,4″-terpyridine, H_2_fum—fumaric acid, H_2_mal—malic acid, Scheme 5) [121]. Both isomeric 2-pytpy and 4-pytpy exhibited emission at 375 nm under excitation at 310 nm. Under the same excitation MOFs demonstrated similar emission at 383 nm and 385 nm. An additional lower energy emission at 522 nm was observed for {[Zn(2-pytpy)(fum)]∙H_2_O}_n_. This band might be due to the π···π stacking interactions between the coordination polymer layers.

A series of zinc coordination polymers with polydentate nitrogen ligands–1D chains {[Zn(H_2_btca)_2_(4-bpt)]∙H_2_O}_n_, 2D layered networks [Zn_2_(btca)(4-PyBIm)(H_2_O)]_n_ and 3D MOF {[Zn_1.5_(Hbtca)(4-pytpy)]∙H_2_O}_n_ (H_4_btca—1,1′-biphenyl-2,3,3′,5′-tetracarboxylic acid, 4-PyBIm—2-(4-pyridyl)benzimidazole, Scheme 5) was reported in [122]. A mixed ligand-centered and LMCT mechanism was suggested for the photoluminescence of these coordination polymers, their photophysical characteristics are given in Table 3.

### 5.2. Carbazole Derivatives

Cheng et al. synthesized Zn^II^-MOFs based on 3,6-bis(imidazole-1-yl)carbazole (3,6-bimcz, Scheme 6) and carboxylate ligands [123]. The free 3,6-bimcz ligand shows emission at 440 nm under excitation at 336 nm. [Zn(bdc)(3,6-bimcz)]_n_ demonstrated an emission maximum at 401 nm upon excitation at 339 nm, {[Zn(p-pda)(3,6-bimcz)]∙1.5H_2_O}_n_ (p-H_2_pda—p-phenylenediacetic acid, Scheme 6) showed an emission band at 403 nm upon excitation at 319 nm. Blue shifts of the emission maxima compared to the free ligand suggested LMCT nature of these bands. In case of {[Zn(bpda)(3,6-bimcz)]∙0.25H_2_O}_n_ (H_2_bpda—benzophenone-4,4′-dicarboxylic acid, Scheme 6) the emission peak at 528 nm under excitation at 428 nm was red-shifted, indicative of MLCT mechanism. The photocatalytic activity of these compounds were evaluated by the degradation of methylene blue (MB) dye in water under irradiation by a 500 W Xe lamp. For [Zn(bdc)(3,6-bimcz)]_n_ and {[Zn(bpda)(3,6-bimcz)]∙0.25H_2_O}_n_ it took 6 h to decompose about 42% and 73% of MB, respectively. However, {[Zn(p-pda)(3,6-bimcz)]∙1.5H_2_O}_n_ demonstrated higher activity in degradation of MB and ca. 95% of MB was degraded in about 6.5 h under UV light irradiation in presence of this coordination polymer.

The same group [124] prepared Zn^II^ with bis-imidazolyl derivatives, 2-amino-4,4′-bis(imidazol-1-yl)bibenzene (abimb), 2-nitro-4,4′-bis(imidazol-1-yl)bibenzene (nbimb), bis(4-(imidazol-1-yl)phenyl)amine (bimpa), isophthalic (H_2_ipa) and 5-bromoisophthalic (H_2_Br-ipa) acids were used as co-ligands (Scheme 6). The photoluminescent properties of the coordination polymers and the free ligands were investigated in the solid state, the photophysical characteristics are given in Table 3. Among the synthesized coordination polymers, {[Zn(Br-ipa)(abimb)]_2_∙0.5H_2_O}_n_ showed the highest photoactivity in MB decomposition. After 5 h of UV irradiation 84.8% of MB were degraded in presence of {[Zn(Br-ipa)(abimb)]_2_∙0.5H_2_O}_n_ [124].

Cheng et al. prepared two Cd^II^ coordination polymers based on 3,6-bis(imidazol-1-yl)carbazole [125]. In the solid state [Cd(NH_2_bdc)(3,6-bimcz)]_n_ exhibited photoluminescent emission at 440 nm upon excitation at 313 nm and [Cd(1,4-ndc)(3,6-bimcz)]_n_ showed emission maximum at 429 nm under excitation at 318 nm. Degradation of MB was also studied. In the presence of [Cd(NH_2_bdc)(3,6-bimcz)]_n_ 82% of MB decomposed after 5 h of UV irradiation. Under the same conditions, only 69.2% of MB were degraded in the presence of [Cd(1,4-ndc)(3,6-bimcz)]_n_.

Du et al. prepared a Cd^II^-MOF {[Cd_3_(cpczdc)_2_(H_2_O)_5_]∙4H_2_O}_n_ based on 9-(4-carboxyphenyl)-9H-carbazole-3,6-dicarboxylic acid (H_3_cpczdc, Scheme 6) [126]. The free ligand showed emission at ~420 nm upon excitation at 356 nm. Upon excitation at 366 nm, Cd-MOF displayed the same emission maximum but it had lower intensity compared to the ligand. In addition, this Cd-MOF showed a “turn-off” luminescent response to Cu^2+^ ions (4 × 10^−4^ M) and nitrobenzene (6 × 10^−4^ M). XPS measurements confirmed the interaction between Cu^2+^ ions and Cd-MOF and electron transfer from the organic ligand to vacant Cu^2+^ orbitals was assumed to be responsible for the luminescence quenching.

Hou et al. prepared a novel MOF {[Zn(Hcpczdc)(4,4′-bpy)_0.5_(H_2_O)]·2H_2_O}_n_, which demonstrated emission at 512 nm upon excitation at 352 nm. MOF suspension in water demonstrated a Stokes shift of 136 nm (λ_em_ = 433 nm, λ_ex_ = 297 nm) and 32% quantum yield. The luminescence intensity decreased by 96% in the presence of 94 µM of Pb^2+^ ions. Uranyl ions UO_2_^2+^ could be also detected, 94% quenching efficiency was reached at UO_2_^2+^ concentration of 81 µM [127]. Luminescence quenching in the presence of Pb^2+^ ions was attributed to coordination of these ions to the free carboxylic groups in the structure of the network. In case of uranyl ions, an overlap between the UV–Vis absorption band of these ions and the emission band of MOF was observed, suggesting a resonance energy transfer mechanism of luminescence quenching.

### 5.3. Bis(imidazol-1-yl)arenes

Li et al. synthesized a number MOFs based on anthracene chromophore 9,10-bis(1H-imidazol-1-yl)anthracene (dia, Table 3) [128]. The free dia ligand showed emission at 474 nm upon excitation at 370 nm. The photophysical characteristics of the prepared Cd-MOFs are given in Table 3. Furthermore, these MOFs could detect nitroaromatic compounds such as nitrobenzene (NB) and 2,4,6-trinitrophenol (TNP) via the luminescence quenching effect, which was attributed to electron transfer between from the network to electron-deficient nitro-compounds, confirmed by cyclic voltammetry studies.

Cd-MOF {[Cd(tzmb)(1,4-bimb)_0.5_]∙2.5H_2_O}_n_ (H_2_tzmb—4,4′-(1H-1,2,4-triazol-1-yl)methylene-bis(benzoic acid), 1,4-bimb—1,4-bis(imidazol-1-ylmethyl)benzene, Table 3) showed tzmb-centered emission at 461 nm upon excitation at 280 nm [129]. It demonstrated luminescence quenching in the presence of some nitroaromatic compounds and metal ions: p-nitrotoluene (98.96%), p-nitrophenol (82.86%), p-nitroaniline (85.90%), nitrobenzene (78.74%) and Fe^3+^ (98.43%).

Three Zn^II^-MOFs based on 4,6-di(1H-imidazol-1-yl)-pyrimidine (dipm) and terephthalates (bdc^2-^, NH_2_bdc^2−^, OHbdc^2−^) were reported in [130]. Topology and interpenetration degree of these MOFs were dependent on the type of dicarboxylate used (Figure 7). The free dipm ligand under excitation at 365 nm showed an emission maximum near 475 nm. The solid-state PL spectra of the coordination polymers were investigated upon excitation at 365 nm: {[Zn(dipm)(bdc)]∙CH_3_OH∙H_2_O}_n_ (λ_em_ = 475 nm), {[Zn(dipm)(NH_2_bdc)]∙2H_2_O}_n_ (λ_em_ = 525 nm), [Zn(dipm)(OHbdc)]_n_ (λ_em_ = 465 nm), their emission was attributed to mixed intraligand and ligand-to-ligand electron transitions.

Vasylevsyi et al. prepared Zn^II^ and Cd^II^ coordination polymers based on 9,10-di(1H-imidazol-1-yl)-anthracene ligand (dia) [131]. The solid-state PL spectrum of the free ligand showed an emission maximum at 433 nm upon excitation at 375 nm with QY = 28%. The excitation and emission maxima of the coordination polymers (Table 3) corresponded well to those of the free ligand and were thus assigned to intra-ligand n-π* and π-π* transitions. Compound {[Zn(µ_2_-dia)_2_(CF_3_CO_2_)_2_]⋅2dioxane}_n_ showed luminescent response to 4,6-dinitropyrogallol with a blue-shift ca. 50 nm.

Xin et al. prepared three zinc coordination polymers based on 1,2-phenylenediacetic acid (H_2_phda) and auxiliary N,N’-bitopic ligands [132]. {[Zn_2_(phda)_2_(4-bpo)_2_]∙2H_2_O}_n_ showed an emission peak at 383 nm and a weak peak near 471 nm upon excitation at 316 nm. [Zn(phda)(4-bpt)]_n_ displayed an emission maximum at 371 nm upon excitation at 310 nm. For {[Zn(phda)(bib)]∙H_2_O}_n_ the emission maxima were observed at 403 nm (λ_ex_ = 340 nm) and 377 nm (λ_ex_ = 305 nm). The fluorescence of the dehydrated [Zn(phda)(bib)]_n_ compound exhibited emission at 395 nm (λ_ex_ = 340 nm) and 335 nm (λ_ex_ = 305 nm). These blue-shifts were attributed to the decreased interchromophore interactions between the interpenetrated nets [132].

{[Zn_2_(1,4-ndc)_2_(3-abpt)]∙2DMF}_n_ (3-abpt—4-amino-3,5-bis(3-pyridyl)-1,2,4-triazole, Table 3) and {[Cd(1,4-ndc)(3-abit)]∙H_2_O}_n_ (3-abit—4-amino-3,5-bis(imidazol-1-ylmethyl)-1,2,4-triazole, Table 3) demonstrated emission at 419 nm (λ_ex_ = 343 nm) and 428 nm (λ_ex_ = 350 nm), respectively [133]. The photoluminescence of MOFs was attributed to ligand-centered excitations. Both MOFs could detect Fe^3+^ and Al^3+^ ions in aqueous solutions. The presence of Fe^3+^ ions lead to the complete quenching of the luminescence, while Al^3+^ caused a significant luminescence enhancing effect. In addition, nitrobenzene and 2,4,6-trinitrophenol could be detected through luminescence quenching effect.

### 5.4. Bis(1,2,4-triazol-1-yl)arenes

Jin et al. prepared a MOF {[Zn(btrbdc)]∙2.7DMF}_n_ based on 2,5-bis(1,2,4-triazol-1-yl)terephtalic acid (H_2_trbdc, Table 3) [134]. The MOF adopted a 3D unimodal 4-c CdSO_4_ topology. A free bis(triazolyl) ligand upon excitation at 280 nm showed an emission peak at 380 nm, while the emission band of Zn-MOF was observed at 432 nm under the same excitation, a notable red-shift was attributed to LMCT. This MOF had a luminescent turn-off response on Fe^3+^ ions, luminescence lifetime was also reduced from 362.17 ns to 44.63 ns with the increase of Fe^3+^ concentration, these changes were attributed to weak binding of Fe^3^+ ions by N/O centers, confirmed by XPS studies. A significant quenching effect was also observed in the presence of TNP, ascribed to charge transfer from the framework to LUMO of TNP, supported by DFT calculations. Additionally, a photocatalytic activity of this MOF was observed in the photo-degradation of methyl violet (MV) and Rhodamine B (RhB) in an aqueous solution under UV irradiation (250 W Hg lamp). As a result, 61.6% of MV and 88.8% of RhB degraded in the presence of Zn-MOF.

Wang et al. prepared a 2D Cd^II^-framework based on 4,4′-bis(1,2,4-triazolyl-1-yl)biphenyl (btb, Table 3) [135]. The free tptz ligand in DMF showed emission at 505 nm upon excitation at 360 nm. In the solid-state tptz exhibited blue emission at 450 nm upon excitation at 350 nm. [Cd(btb)(H_2_O)_2_(HCOOH)(ipa)_2_]_n_ showed emission at 450 nm (λ_ex_ = 360 nm) in DMF suspension and at 550 nm (λ_ex_ = 350 nm) in the solid-state. This MOF demonstrated detection of electron-deficient compounds, such as nitrobenzene and p-dinitrobenzene with a quenching efficiency of 99.2% and 99.998%, respectively.

Two copper(II) coordination polymers based on btb and pyridine-2,5-dicarboxylic acid (H_2_pydc) or 3-nitrophthalic acid (H_2_nph) as co-ligands were prepared by Wang et al. [136]. The btb ligand exhibited a broad emission band at 439 nm upon excitation at 380 nm. Both of the coordination polymers showed strong emission peaks at 410 nm (λ_ex_ = 360 nm) for [Cu(btb)(pydc)(H_2_O)]_n_ and 333 nm (λ_ex_ = 210 nm) for [Cu(btb)_0.5_(nph)(H_2_O)]_n_.

Nickel(II) 2D coordination polymer with the same btb ligand was synthesized by the same research group [137]. An intense emission of [Ni_3_(btb)_4_(nbta)_2_(H_2_O)_4_]_n_ (H_3_nbta—5-nitro-1,2,3-benzenetricarboxylic acid) was observed at 418 nm upon excitation at 230 nm.

Copper(I) and silver(I) coordination polymers, [Cu_2_(btb)(CN)_2_]_n_ and [Ag_2_(btb)(muco)]_n_ demonstrated broad peaks at 403 nm (λ_ex_ = 300 nm) and 388 nm (λ_ex_ = 360 nm), respectively [138]. Both coordination polymers exhibited catalytic activity in methyl orange (MO) decomposition. In the presence of Cu-CP and Ag-CP after 150 min MO decomposed by 68.9% and 96.6%, respectively.

Copper(I) cyanide coordination polymers, [Cu(btb)(CN)]_n_, [Cu_2_(bdmbib)(CN)_2_]_n_, [Cu_2_(bmbip)(CN)_2_]_n_ and [Cu_2_(bdmbip)(CN)_2_]_n_ (bdmbib—1,4-bis(5,6-dimethylbenzimidazol-1-yl)butane, bmbip—1,3-bis(2-methylbenzimidazol-1-yl)propane, bdmbip—1,5-bis(5,6-dimethylbenzimidazol-1-yl)pentane) were prepared by Zhang [139]. The photophysical characteristics are given in Table 3. It is interesting to note that both the free bdmbip ligands and the coordination polymer [Cu_2_(bdmbip)(CN)_2_]_n_ demonstrated no emission. All coordination polymers demonstrated catalytic activity in Congo red decomposition: 83.5% for [Cu(btb)(CN)]_n_, 96.8% for [Cu_2_(bdmbib)(CN)_2_]_n_, 84.1% for [Cu_2_(bmbip)(CN)_2_]_n_ and 72.2% for [Cu_2_(bdmbip)(CN)_2_]_n_ [139].

### 5.5. Benzimidazole Derivatives

Zong et al. prepared a Cd-MOF [Cd(tmb)(bbibp)]_n_ based on 4,4′-bis(benzimidazo-1-yl)biphenyl (bbibp) and (1*H*-1,2,4-triazol-1-yl)methylenebis(benzoic acid) (H_2_tzmb), which represented a two-fold interpenetrated unimodal 4-connected 3D frameworks with a (6^5^∙8) topology [140]. The free bbibp ligand at the solid-state showed emission peak at 400 nm upon excitation at 275 nm, H_2_tmb demonstrated emission band at 420 nm upon excitation at 330 nm. However, when the ligands were coordinated by Cd^2+^ the main emission peak of the MOF appeared at 360 nm upon excitation at 285 nm and was assigned to ligand-centered electron transitions. Ion detection by this Cd-MOF was also studied. The sample of Cd-MOF was immersed in different aqueous solutions containing 0.01 M K_2_CO_3_, K_2_Cr_2_O_7_, K_2_SO_4_, KSCN, K_2_CrO_4_, KCl, KClO_3_, KIO_3_, KNO_3_, KOH or KH_2_PO_4._ The MOF suspensions were sonicated in the dark for 30 min. As a result, Cr_2_O_7_^2−^ and CrO_4_^2−^ anions exhibited luminescence intensity quenching effect. The same effect was observed in detection of Fe^3+^ ion (0.01 M) in an aqueous media. Additionally, detection of organic solvents was investigated. The as-synthesized sample was dispersed in different organic solvents, including MeOH, EtOH, MeCN, DMA, DMF, IPA, DCM, NB, THF and benzene. The highest luminescence intensity was observed in DMF, while NB showed an obvious luminescence quenching effect [140]. The luminescence quenching effect was ascribed to photo-induced electron transfer from the excited state of MOF to electron-deficient NB, a favorable HOMO-LUMO gap was confirmed by DFT calculations.

Cd^II^-MOF [Cd(Hcbic)]_n_ (H_3_cbic–1-(4-carboxybenzyl)-1H-benzoimidazole-5,6-dicarboxylic acid, Table 3) with a strong emission peak at 377 nm upon excitation at 320 nm demonstrated a high quenching degree (up to 97.8%) in the presence of Fe^3+^ ions [141]. Blue-shift of the emission band relative to the free ligand allowed to tentatively assign it to LMCT. Luminescence quenching by Fe^3+^ ions was attributed to adsorption of these ions by the coordination polymer, confirmed by ICP analysis.

Wang et al. prepared Cd^II^-MOFs based on 2-pyridin-3-yl-1H-benzoimidazole (3-PyBim) and 2-pyridin-4-yl-1H-benzoimidazole (4-PyBim) [142]. Free 3-PyBim and 4-PyBim ligands under excitation at 280 nm showed two-shoulder emission peaks at 417 nm and 540 nm for 3-PyBim, and 411 nm and 516 nm for 4-PyBim. All of the MOFs demonstrated luminescence in the solid-state upon excitation at 280 nm: [CdI_2_(3-PyBim)](H_2_O)_3_]_n_ (λ_em_ = 404 nm), [Cd(SO_4_)(3-PyBim)(H_2_O)_4_]_n_ (λ_em_ = 413 nm), [CdCl_2_(4-PyBim)_2_(H_2_O)_2_]_n_ (λ_em_ = 451 nm), [CdBr_2_(4-PyBim)_2_(H_2_O)_2_]_n_ (λ_em_ = 444, 540 nm) and [CdI_2_(4-PyBim)_2_(H_2_O)_2_]_n_ (λ_em_ = 435 nm).

Three coordination polymers, {[Zn_2_(tzmb)_2_(bim)]∙2DMA∙EtOH}_n_, {[Zn_2_(tzmb)_2_(4,4′-bipb)∙2DMA⋅EtOH]}_n_ and {[Cd(tzmb)(bbibp)]∙DMA∙EtOH}_n_ (H_2_tzmb—4,4′-(1H-1,2,4-triazol-1-yl)methylenebis(benzonic acid), bim—benzimidazole, 4,4′-bipb—4,4′-bis(pyrid-4-ly)biphenyl, bbibp—4,4′-bis(benzoimidazol-1-yl)biphenyl, Table 3) were reported in [143], their photophysical characteristics are given in Table 3. The emission bands were assigned to intraligand π*-π or π*-n transitions. Both Zn coordination polymers could detect Fe^3+^ ions in an aqueous solution via luminescence intensity quenching, which was attributed to the overlap between the absorption band of Fe^3+^ and the emission bands of the coordination polymers.

### 5.6. Other N-heterocyclic Derivatives

Dai et al. prepared a series of Zn^II^, Co^II^ and Cd^II^ coordination polymers based on 4-((3-(pyrazine-2-yl)-1H-pyrazol-1-yl)methyl)benzoic acid (Hpypymba), 4-((3-(pyridine-2-yl)-1H-pyrazol-1-yl)methyl)benzoic acid (Hpyznpy) and 2-(3-pyridin-2-yl)-1H-pyrazol-1-yl)acetic acid (Hpypyaa) [144]. Their emission was attributed to intraligand π-π* transitions, the photophysical characteristics of the coordination polymers are shown in Table 3. Fluorescent detection of solvents and metal ions was investigated. Coordination polymer {[Cd_2_(pyznpy)_2_(H_2_O)Cl_2_]∙H_2_O}_n_ demonstrated a quenching effect in the presence of nitrobenzene (1 mM) and in the presence of Fe^3+^ ions (in EtOH solution). The decrease of luminescence intensity by 99.99% was observed at Fe^3+^ concentration of only 9 × 10^−7^ mol/L. Electron transfer from the conduction band of MOF in the excited state to the LUMO of the organic analytes was proposed as a possible quenching mechanism.

Han et al. prepared Cd^II^-MOF based on N-heterocyclic multicarboxylate ligand 3,3′-((6-amino-1,3,5-triazine-2,4-diyl)bis(azanediyl))dibenzoic acid (H_3_atbdc) [145]. The structure of this MOF featured hexagonal hydrophobic channels that allowed to include fused aromatic hydrocarbons (FAH): anthracene, naphthalene and phenanthrene. The MOF showed an emission peak at 397 nm upon excitation at 338 nm. In the presence of FAH the MOF showed luminescence intensity enhancement effect, attributed to π-π and van der Waals interactions between FAH molecules and hydrophobic MOF channels.

Zhao et al. prepared Tb-MOF based on 5-(1H-pyrazol-3-yl)isophthalic acid (H_2_pia) [146]. Free H_2_pia ligand displayed an emission peak at 393 nm upon excitation at 331 nm. The solid-state luminescence spectrum of [Tb(Hpia)(pia)(H_2_O)_2_]_n_ featured four emission peaks of Tb^3+^ at 493, 456, 589 and 624 nm (λ_ex_ = 254 nm) corresponding to the f-f transitions ^5^D_4_–^7^F_6_, ^5^D_4_–^7^F_5_, ^5^D_4_–^7^F_4_ and ^5^D_4_–^7^F_3_, respectively. This Tb-MOF demonstrated a luminescence turn-off effect on phosphate anions in aqueous suspension. When the concentration of PO_4_^3−^ reached 3.27·10^−4^ M, the luminescence intensity at 546 nm (^5^D_4_–^7^F_5_ transition) was quenched almost entirely.

A 3D pillar-layered molybdenum-oxide-based MOF [Zn_2_(tyty)(µ_3_-OH)(MoO_4_)(H_2_O)]_n_ (Htyty–1-(tetrazol-5-yl)-4-(1,2,4-triazol-1-yl)benzene) was reported in [147]. The MOF exhibited emission peak at 409 nm under excitation at 300 nm, while the free Htyty ligand showed emission at 370 nm upon excitation at 284 nm.

Zn-MOF {[Zn(dptmia)]∙2H_2_O·0.5DMF}_n_ (dptmia—(5-(3,5-di-pyridin-4-yl-[1,2,4]triazol-1-ylmethyl)-isophthalic acid) with the emission peak near 430 nm (λ_ex_ = 350 nm) was tested for the sensing of metal ions and organic compounds [148]. It was found that Fe^3+^ ions in the aqueous MOF suspension lead to a considerable luminescence quenching. Among the organic analytes tested, 4-nitrophenol, 2,4-dinitrohenol and 3-nitrophenol showed the most notable quenching effects. Structural changes of MOF observed by PXRD method after its exposure to Fe^3+^ ions, suggested a cation-exchange of Zn^2+^ as a mechanism of luminescence quenching.

Liu et al. prepared a MOF {(Me_2_NH_2_)[Zn_2_(dcpp)(OAc)]∙0.5DMF}_n_ [149]. The free pyridine-based ligand exhibited an emission at 375 nm upon excitation at 312 nm, while the coordination polymer showed a strong emission peak at 428 nm under excitation at 320 nm. Luminescence quenching effect was observed in the presence of nitrobenzene in DMF solution, the emission intensity decreased by 50% at 75 ppm and complete quenching was achieved at 175 ppm of nitrobenzene, the quenching effect was attributed to energy transfer from the ligand to nitrobenzene molecules.

Duan et al. [150] prepared stable Ln-MOFs, two of which {[Eu_4_(pta)_5_(Hpta)_2_(H_2_O)_4_]·9H_2_O}_n_ and {[Tb(Hpta)(C_2_O_4_)]·3H_2_O}_n_ (H_2_pta—2-(4-pyridyl)-terephthalic acid) displayed emission. Eu-MOF upon excitation at 366 nm displayed emission peaks at 581, 593, 613 and 653 nm corresponding to the characteristic ^5^D_0_–^7^F_0_, ^5^D_0_–^7^F_1_, ^5^D_0_–^7^F_2_ and ^5^D_0_–^7^F_4_ Eu^3+^ transitions. Tb-MOF upon excitation at 376 nm exhibited four emission peaks at 490, 546, 585 and 622 nm, which can be ascribed to ^5^D_4_–^7^F_J_ (J = 6–3) transitions of Tb^3+^ ions. Both MOFs could detect Fe^3+^ ions in an aqueous solution with luminescence intensity quenching by 98%. Tb-MOF exhibited a luminescence turn-off effect in the presence of Cr_2_O_7_^2−^ ions with a high quenching efficiency up to 98%. Moreover, Tb-MOF demonstrated a dramatic luminescence quenching effect of 96% in the presence nitrofuran in DMF solution.

Zhang et al. synthesized Cd^II^-MOFs using 2,6-bis(3-(pyrid-3-yl)-1,2,4-triazolyl)pyridine (H_2_bptp) and different aromatic carboxylates [151]. The free H_2_bptp ligand showed an emission maximum at 523 nm upon excitation at 371 nm. The intense emission bands for the coordination polymers were observed at 410 nm (λ_ex_ = 363 nm) for {[Cd_2_(bptp)_2_(H_2_O)_4_]∙3H_2_O}_n_, 464 nm (λ_ex_ = 396 nm) for {[Cd_2_(Hbptp)(btc)(H_2_O)]∙4H_2_O}_n_ and 456 nm (λ_ex_ = 284 nm) for {[Cd_2_(H_2_bptp)(btec)]∙H_2_O}_n_ (H_4_btec—1,2,4,5-benzenetetracarboxylic acid). Blue shift of the emission bands were attributed to metal-ligand coordination interactions. All three compounds demonstrated photocatalytic activity in degradation of MB in an aqueous solution. {[Cd_2_(bptp)_2_(H_2_O)_4_]∙3H_2_O}_n_ and {[Cd_2_(Hbptp)(btc)(H_2_O)]∙4H_2_O}_n_ exhibited much better photocatalytic activity (decomposition rates 87% and 95%, respectively) than {[Cd_2_(H_2_bptp)(btec)]∙H_2_O}_n_ (32%) [151].

Lee L. et al. prepared a Zn^II^-MOF based on 2,4,5-tri(4-pyridyl)imidazole (Htipm) and 3,4-pyridinedicarboxylic acid (3,4-H_2_pydc) as ligands [152]. It demonstrated interesting thermochromic and solvatochromic effects (Figure 8). Compound {[Zn_2_(Htpim)(3,4-pydc)_2_]∙4DMF·4H_2_O}_n_ showed emission at 423 nm under excitation at 380 nm. When this MOF was heated to 100 °C to remove water it displayed an emission peak at 466 nm and after heating at 160 °C (DMF removal) the emission shifted to 515 nm upon excitation at 380 nm.

Zhang et al. prepared MOFs [Cd(*p*-CNPhHIDC)(4,4’-bipy)_0.5_]_n_ and [Zn(*p*-CNPhHIDC)(4,4’-bipy)]_n_ (*p*-CNPhH_3_IDC—2-(4-cyanophenyl)-1H-imidazole-4,5-dicarboxylic acid), which exhibited ligand-centered emission at 441 nm (λ_ex_ = 380 nm) and 457 nm (λ_ex_ = 370 nm), respectively and could be used for Fe^3+^ detection via luminescence quenching [153].

Both linear and nonlinear photophysical properties of zinc(II) metal-organic frameworks based with *trans*-2-(4-pyridyl)-4-vinylbenzoate (pvb) ligands were reported in [154]. Two MOFs of **dia** topology {[Zn(pvb)_2_]⋅DMF}_n_ and [Zn(pvb)_2_]_n_ with 7-fold and 9-fold interpenetration were synthesized and structurally characterized. Single crystal-to-single crystal structural transformation of the first MOF to the second upon removal of DMF was observed. The intraligand emission wavelength was dependent on the presence of solvent molecules and was red-shifted for [Zn(pvb)_2_]_n_ owing to increased π-π interactions (λ_em_ = 478 nm vs. λ_em_ = 460 nm for {[Zn(pvb)_2_]⋅DMF}_n_, Table 3). Nonlinear optical properties were characterized by measuring the second harmonic generation (SGH) and two-photon photoluminescence (2PPL). Desolvated MOF demonstrated a much stronger SGH response compared to the solvated product, 2PPL intensity was also approximately 14 times stronger for the desolvated MOF.

## 6. Ligands with BODIPY-Type Fluorophores

Boron-dipyrromethene or BODIPY derivatives are notable for their uniquely small Stokes shift, high, environment-independent fluorescence quantum yields, often approaching 100% even in water, sharp excitation and emission peaks contributing to overall brightness [155,156,157,158]. The combination of these qualities makes BODIPY fluorophores promising for the construction of highly luminescent coordination polymers, some of which were reviewed in 2016 [159].

Zhou et al. prepared two pillar-layered MOFs {[Zn_3_(L^1^)_2_(btc)_2_(H_2_O)]·guests}_n_, {[Cd(L^1^)(bdc)]·guests}_n_, with 2,6-di(4-pyridyl)-substituted BODIPY (L^1^, Table 4) as struts linking {Zn_3_(btc)_2_} or {Cd(bdc)} layers into 3D structures (Figure 9) [160]. Both MOFs exhibited similar emission spectra upon excitation at 512 nm in the solid state and emitted strong green light centered at 540 nm, similar to ligand L^1^ in solution (Table 4). The emission band was attributed to S_1_→S_0_ transition of the BODIPY fragment.

Similar pyridine-functionalized ligand L^2^ (Table 4) was employed by Yang et al. for building two MOFs [Cd_2_(L^2^)_2_(bpdc)_2_]_n_, CCNU-11 and [Cd_2_(L^2^)_2_(sdb)_2_]_n_, CCNU-12 (H_2_bpdc—4,4′-biphenyldicarboxylic acid, H_2_sdb—4,4′-sulfonyldibenzoic acid) with the identical topology [161]. Upon excitation at 440 nm, CCNU-11 or CCNU-12 aqueous suspensions (1 g/L) demonstrated emission bands centred at 624 nm (Table 4). These solids were employed as photocatalysts for the H_2_ evolution in the presence of [Co(2,2′-bpy)_3_]Cl_2_) co-catalyst.

In another work by the same group, composite material of coordination polymer {[Zn_2_(L^2^)_2_(bpdc)_2_]·H_2_O}_n_ and Pt nanoparticles was used for efficient hydrogen production with the rate of 4680 μmol·g^−1^·h^−1^ one of the highest among MOF photocatalysts [162].

Li et al. successfully synthesized five coordination polymers with dicarboxylate-functionalized BODIPY 2,6-dicarboxyl-1,3,5,7-tetramethyl-8-phenyl-4,4-difluoroboradiazaindacene (H_2_L^3^, Table 4), namely, {[Zn_2_(L^3^)_2_(bpp)]·2H_2_O·2EtOH}_n_, {[Cd_2_(L^3^)_2_(bpp)]·2H_2_O·EtOH}_n_, {[Cd_2_(L^3^)(bpe)_3_(NO_3_)_2_]·2H_2_O·DMF·EtOH}_n_, {[Cd(L^3^)(bpe)_0.5_(DMF)(H_2_O)]}_n_ and {[Cd(L^3^)(bpe)_0.5_]·1.5H_2_O·DMF}_n_ (bpp—1,3-bi(4-pyridyl)propane, bpe—1,2-bi(4-pyridyl)ethane) [163]. Spectroscopic investigations showed that in the case of {[Cd_2_(L^3^)_2_(bpp)]·2H_2_O·EtOH}_n_ an uncommon J-dimer absorption band was observed at λ_max_ = 705 nm with a long tail into the NIR region at room temperature. Low quantum yields (less than 1%) were explained by the formation of dimers by BODIPY moieties and concentration quenching. Interestingly enough, QY of the coordination polymer {[Cd_2_(L)(bpe)_3_(NO_3_)_2_]·2H_2_O·DMF·EtOH}_n_ is twice as high as of other complexes (1.7%), since the structure of this framework did not allow the formation of dimers.

Two BODIPY-based MOFs are reported in [164]. They were formulated as [Zn_2_(dbtcpb)(L^2^)], BOB-MOF and [Zn(Zntpcp)(L^2^)], BOP-MOF (H_4_dbtcpb—3,6-dibromo-1,2,4,5-tetrakis(4-carboxyphenyl)benzene, H_4_tpcp—Zn-tetraphenylcarboxyporphyrin). BOP-MOF was found to be capable of light harvesting across the entire visible spectrum, while producing luminescence centered near 667 nm upon excitation at 520 nm (Figure 10). BOB-MOF showed the emission maximum at 596 nm upon excitation at the same wavelength (Table 4).

Coordination polymer based on BODIPY-Ag-metallomacrocycle {[Ag_2_(L^4^)_2_(MeOH)](AsF_6_)_2_}_n_ (the structure of L^4^ is shown in Table 4) was reported in [165]. It produced two emission bands centered at 750 and 790 nm upon excitation at 530 nm. The bands are red-shifted in comparison to the free BODIPY-ligand, which is consistent with the data for other BODIPY-derived coordination polymers.

## 7. Naphthalene Diimide Derivatives

Naphthalene diimide (NDI) are planar π-electron deficient redox-active molecules, which makes them a popular platform for the construction of coordination polymers with ligand-based luminescence [166,167].

Two novel 2D coordination polymers, {[Cd(bipatc)_0.5_(bib)(H_2_O)]·3.5H_2_O}_n_ and {[Cd_2_(bipatc)(bibp)(H_2_O)]·H_2_O}_n_ (H_4_bipatc—5,5′-(1,3,6,8-tetraoxobenzo[*lmn*][3,8]phenanthrolin-2-7-diyl)bis-1,3-benzenedicarboxylic acid, Table 5, bib—1,4-bis(imidazol-1-yl)benzene, bibp—4,4′-bis(imidazol-1-yl)biphenyl) are presented in [168]. Both compounds exhibited strong luminescence with the maxima near 325 and 350 nm, respectively. Furthermore, compounds displayed a sensing ability for nitroaromatics in DMF suspensions, acetone and Fe^3+^/Hg^2+^ ions in EtOH through luminescence quenching. Effect of the analyte concentration on the luminescence was surveyed and the detection limits were determined.

NDI derivatives are quite often related to the design of photochromic materials. Wei et al. synthesized Zn-based MOF {[Zn(bipatc)_0.5_(4,4′-bpy)(H_2_O)]·3.5H_2_O}_n_, FJU-34 [169]. Upon excitation at 390 nm it produced a broad emissive band centered at 550 nm. On increasing the irradiation time, luminescence intensity decreased gradually, reaching about 71% of the initial value after one hour. It corresponds with the change of color of the sample from yellow to brown. This is due to the NDI’s ability to form radicals upon irradiation, which was confirmed by ESR spectroscopy [169].

The same group reported three Cd-MOFs built from NDI-pyrazolate ligand (PzNDI, Table 5) and different dicarboxylic acids, [Cd(bdc)(PzNDI)]_n_ (FJU-67), [Cd(NH_2_bdc)(PzNDI)]_n_ (FJU-68, H_2_NH_2_bdc—2-aminoterephthalic acid) and {[Cd(2,6-ndc)(PzNDI)]·2H_2_O}_n_ (FJU-69) [170]. Coordination polymers FJU-67 and FJU-69 demonstrated photochromic behavior due to the photochemical radical generation. While the free ligand showed an emission peak at 590 nm upon irradiation at 345 nm, the complexes exhibited slightly blue-shifted (by 20 nm and 13 nm) emissions bands, attributed to MLCT. As a result that the photoresponsive ability of FJU-67 was found to be slightly higher, the authors suggested that the interpenetrating mode plays a definite role in influencing the efficiency of the photoinduced electron transfer reactions, resulting in different degrees of photoresponse [170].

Another photochromic MOF was reported in [171]. It resembled a Zn-based structure that is built with the NDI-derived ligand, which incorporated pyridine donors (isondi—N,N′-di(4-pyridylacylamino)-1,4,5,8-naphthalenediimide, Table 5). It was formulated as {[Zn(2,6-ndc)(isondi)(H_2_O)_2_]·2H_2_O}_n_ and underwent a color change from dark red to brown upon UV-Vis irradiation. It also exhibited a broad emission band near 660 nm with an absolute quantum yield of 0.02% (Table 5). The fluorescence was quenched after prolonged irradiation.

A series of nine MOFs, namely {[Mn(L^5^)(NCS)_2_(MeOH)_2_]·2MeOH}_n_, {[Mn(L^6^)_2_(NCS)_2_(MeOH)_2_]}_n_, {[Mn(L^7^)_2_(NCS)_2_]·0.5MeOH·4.5H_2_O}_n_, {[Cd(L^5^)I_2_]·3H_2_O}_n_, {[Cd(L^6^)I_2_]}_n_, {[Cd(dpmni)(NCS)_2_]·2.5H_2_O}_n_, {[Co(L^5^)_2_(NCS)_2_]·4H_2_O}_n_, {[Co(L^6^)_2_(NCS)_2_]·CHCl_3_}_n_ and {[Co(dpmni)_2_(NCS)_2_]·3H2O}_n_ (L^5^—N,N-bis(4-pyridylmethyl)pyromellitic diimide, L^6^—N,N’-bis(3-pyridylmethyl)pyromellitic diimide, dpmni—N,N’-bis(4-pyridylmethyl)naphthalene diimide, Table 5) was reported in [172]. The luminescent properties were studied only for {[Cd(L^6^)I_2_]}_n_ and {[Cd(dpmni)(NCS)_2_]·2.5H_2_O}_n_. These coordination polymers displayed broad ligand-centered emission bands at 437 and 575 nm, with excitation at 300 and 465 nm, respectively (Table 5). The red-shift was explained by increased rigidity of the ligands upon coordination to the metal centers.

Three photochromic MOFs [Zn_2_(dpmni)(tdc)_2_]_n_, [Zn(dpmni)(dpdc)]_n_,[Zn(dpmni)(Hbtc)]_n_ were reported in [173]. After irradiation, they demonstrated additional absorption peaks centered near 485, 620 and 765 nm, respectively, in addition to the original absorption at 450 nm. The photochromic behavior was attributed to the photoinduced radical generation of dpmni ligands.

The first synthesis of a coordination polymer with amino-acid functionalized NDI ligand (GlyNDI, Table 5) was reported in [174]. Compound [Zn(GlyNDI)(dmf)_2_]_n_ had a 1D zig-zag chain structure and exhibited strong fluorescence centered at 458 nm upon excitation at 320 nm. The mechanism of the luminescence was described as metal-to-ligand charge transfer (MLCT) and ligand-to-ligand charge transfer (LLCT), supported by intermolecular π-π interactions.

Xu et al. prepared a range of coordination networks with different metal ions, {[Zn_2_(bindi)(dma)_2_]·2DMA}_n_, {[Cd(H_4_bindi)(H_2_bindi)]·4DMF}_n_, {[Ca_2_(Hbindi)Cl(dma)_3_]·DMA}_n_, {[Ba_4_(bindi)_2_(dmf)_7_]·7DMF} (H_4_bindi—N,N′-bis(5-isophthalic acid)naphthalenediimide, Table 5) [175]. All of the complexes were photochromic and their photosensitivity increased in the order Ba(II) < Ca(II) < Cd(II) < Zn(II), consistent with the order of electronegativity for these ions. Photoluminescent properties were also surveyed. The emission was centered at 585 nm for Cd and Zn compound when excited at 280 nm. For Ca compound the emission maximum was at 499 nm and for Ba it was at 467 nm. In addition, these compounds exhibited a color change upon exposure to nitrite ions and different solvents (Figure 11).

Fu et al. synthesized two cadmium-based coordination polymers, [Cd(dpmni)(ipa)]_n_ and [Cd(dpmni)(bdc)]_n_ [176]. The first compound exhibited three emission peaks at 424, 452 and 475 nm upon excitation at 380 nm. Multiply emission peaks were attributed to ligand-centered excitation and MLCT. The second coordination polymer exhibited emission centered at 484 nm upon excitation at 350 nm. Due to their photochromism, upon irradiation with UV light, the intensities of their luminescence decreased to 52% and 61% of their original values.

Photocontrolled luminescence is a fairly usual occurrence in the studies regarding photochromic materials. For example, three coordination polymers, [Cd(isondi)(2,6-ndc)(H_2_O)_2_]_n_, [Cd(isondi)(ipa)(dmf)]_n_ and [Cd_2_(isondi)_2_(bdc)_0.5_(HCOO)_2_]_n_ demonstrated essentially complete fluorescence quenching after brief UV irradiation [177]. The luminescence was recorded after the samples were kept in the dark for some time. [Cd(IsoNDI)(2,6-ndc)(H_2_O)_2_]_n_ showed two emission peaks at 628 and 403 nm upon excitation at 380 nm, [Cd(IsoNDI)(ipa)(dmf)]_n_ demonstrated two peaks at 418 nm and 438 nm (excitation at 375 nm). The emission spectrum of [Cd_2_(IsoNDI)_2_(bdc)_0.5_(HCOO)_2_]_n_ contained three peaks at 534, 570 and 612 nm (excitation at 390 nm) [177]. All emission peaks were attributed to ligand-centered excitation.

Another material with a photocontrolled tunable luminescence was reported in [178]. Being a host-guest compound with polyoxometalates as guests, it had an overall formula [C_88_H_100_F_4_W_24_N_16_O_96_P_2_Zn_3_]_n_ and underwent color change upon UV-irradiation. It emitted light with a maximum near 450 nm when excited at 350 nm. It was found that this compound could serve as an effective photocatalyst for selective oxidation of benzylic alcohols [178].

Liu et al. reported three isostructural 1D coordination polymers with the formula [Zn(dpndi)X_2_]_n_, where X = Cl^−^, Br^-^ or I^−^ [179]. All compounds showed emission at around 595 nm upon excitation at 490 nm, which was completely quenched under UV light for [Zn(dpndi)Cl_2_]_n_ and [Zn(dpndi)Br_2_]_n_, but no obvious changes were observed for [Zn(dpndi)I_2_]_n_. This difference was explained by the strength of lone pair-π interactions being the highest for the structure with iodine ions, thus, the π-acidity of dpndi moiety being the lowest one.

The same group reported the coordination polymer {[Cd(dpmni)_2_(ClO_4_)_2_]·4CHCl_3_}_n_, which exhibited photochromic behavior upon exposure to UV light and emission centered at 467 nm upon excitation at 340 nm [180].

In the work of Shang et al., MOF [Zn(AlaNDI)]_n_ was synthesized with the use of chiral R- or S-alanine-functionalized NDI ligands [181]. The resulting structure displayed a number of interesting properties. Among them is an intense luminescence centered at 475 nm (360 nm excitation) which could be quenched by Naproxen—a non-steroidal anti-inflammatory drug, which is chiral. Interestingly, if non-racemic MOF is used, enantiodifferentiating fluorescence sensing is possible.

One of the five compounds studied in [182] was a 1D coordination polymer with the formula {[Cd_2_(3-imntd)_2_I_4_]·2H_2_O}_n_ (3-imntd—N,N′-bis(3-imidazol-1-yl-propyl)naphthalene diimide, Table 5). It demonstrated two-centered ligand-based emission at 410 nm and 590 nm upon excitation at 309 nm. Since both of the emission bands matched the same bands of the free ligand 3-imntd, they were assigned to intra-ligand emissions. The longer-wavelength band was assigned to the transition between imidazole and NDI with perpendicular orientation, while the shorter-wavelength band was assigned to the same transition with the coplanar orientation of the groups [182].

Dhankhar et al. reported a pillar-layered framework [Zn_2_(NH_2_bdc)_2_(dpndi)]_n_ [183]. Water suspension of this compound exhibited strong ligand based (π*-n and π*-π) emission at 430 nm under 330 nm light. This suspension was screened for the ability to detect the presence of nitroaromatic compounds (NAC) in solution (Figure 12). It was found that all NAC cause fluorescence quenching, but 2,4,6-trinitrophenol (TNP) had the strongest effect, quenching nearly 92% of the initial intensity and shifting the maximum to 460 nm, which could be used for selective detection of TNP. DFT studies suggested that both photo-induced electron transfer and Förster resonance energy transfer processes are responsible for the luminescence quenching.

Coordination polymer {[Zn_2_(hfipbb)_2_(dpndi)]⋅8DMF}_n_ (H2hfipbb—4,4′-(hexafluoroisopropylidene)bis(benzoate)) was reported in [184]. It exhibited strong LMCT emission with the maximum at 415 nm upon excitation at 360 nm. Redox and photochromic properties have also been surveyed.

Coordination polymer [Zn(tzndi)(dmf)_2_]_n_, NBU-3 (tzndi—N,N′-bis(tetrazol-5-yl)naphthalene diimide) prepared in [185] displayed both photo- and electrochromic properties. Its luminescence upon excitation at 322 nm was centered at 468 nm and attributed to LLCT and/or LMCT.

An interesting work from Kitagawa et al. [186], in which MOF [Zn_2_(bdc)_2_(dpndi)]_n_ was prepared for the first time, shows how useful the NDI-based frameworks can be. This interpenetrated framework was able to incorporate molecules of various organic volatiles, which was accompanied by changes in crystal structure and shifts of luminescence maximum to a specific position for each compound (Figure 13). Furthermore, quantum yield and luminescence lifetime were also affected. For a full range of compounds see Table 5.

Deng et al. reported the synthesis of complex [Cd(3-pmntd)_2_(NO_3_)_2_]_n_ (3-pmntd—N,N′-bis(3-pyridylmethyl)naphthalene diimide), a luminescent material, which upon excitation at 447 nm exhibited broad emission band centered at 557 nm [187]. The emission band was structured compared to a wide band of the free ligand, which suggested that charge-transfer transitions contributed to the emission along with the ligand-centered excitation.

Yang et al. prepared NDI-based Ag-MOF, which was formulated as [Ag(2,6-ndc)_0.5_(dpndi)_0.5_(H_2_O)]_n_ [188]. It is a stable supramolecular material with good electrical conductivity and interesting luminescent properties. Its MLCT emission was centered at 370 nm (excitation at 320 nm) and experienced a strong turn-on effect upon exposure to dichloromethane, while in other organic solvents the luminescence was quenched. This makes the aforementioned coordination polymer a good candidate for selective dichloromethane probes.

Magnesium is an ion much rarer encountered compared to Zn and Cd in luminescent MOFs. There are, however, some examples. Mallick et al. prepared Mg-based MOF, which was determined to be [Mg_2_(bindi)_2_(dmf)_2_(H_2_O)]_n_ [189]. Dry samples showed strong emission band centered at 570 nm upon excitation at 515 nm, but upon exposure to solvents a gradual blue-shift consistent with the increase in polarity of the solvent occurred (Figure 14), bringing the maximum to 625 nm at the highest solvent polarity (EtOH). The luminescence was quenched in presence of both aliphatic and aromatic amines. Interestingly, sensing experiments were conducted with the as-synthesized MOF in the solid state, whereas in most works the studies are carried out with MOF suspensions in some solvent.

Calcium-based MOF {[Ca_2_(bindi)(dmf)_4_]·2DMF}_n_ with excellent thermal stability and photochromic properties was described in [190]. Pristine sample showed 575 nm centered emission under 505 nm excitation light, which was attributed to the LLCT and/or LMCT.

An unprecedented linear correlation between the full width at half maximum (FWHM) of PL bands and the π-π distances between aromatic ligands was demonstrated in a work utilizing γ-aminobutyric acid-functionalized NDI [191]. Sixteen MOFs were prepared and characterized. They were formulated as follows: [Li_2_(GABA-NDI)]_n_, [Na(HGABA-NDI)]_n_, [K(HGABA-NDI)]_n_, [Ca(GABA-NDI)(MeOH)(dmf)]_n_, [Ca(GABA-NDI)(MeOH)]_n_, [Cd(GABA-NDI)(MeOH)_2_]_n_, [Mg(GABA-NDI)(MeOH)_2_]_n_, [Mn(GABA-NDI)(MeOH)_2_]_n_, [Ni(GABA-NDI)(MeOH)_2_]_n_, [Co(GABA-NDI)(EtOH)_2_]_n_, [Zn(GABA-NDI)(dmf)]_n_, [La_2_(GABA-NDI)_3_(dmf)]_n_, [Eu_2_(GABA-NDI)_3_(dmf)]_n_, [Tb_2_(GABA-NDI)_3_(dmf)]_n_, [Sm_2_(GABA-NDI)_3_(dmf)]_n_, [Gd_2_(GABA-NDI)_3_(dmf)]_n_. Their PL characteristics are listed in Table 5.

## 8. Derivatives of 4,4′,4′′-Nitrilotribenzoic Acid

4,4′,4′′-Nitrilotribenzoic acid (H_3_ntb, Scheme 7) has an extended π-electron system, which ensures pronounced luminescent properties of this tritopic ligand and its coordination compounds.

A series of zinc coordination polymers based on 4,4′,4′′-nitrilotribenzoic acid and several co-ligands was reported in [192]. Coordination polymers {(Me_2_NH_2_)_2_[Zn_2_(ntb)_2_(py)_2_]·DMF·2H_2_O}_n_, {[Zn_2_(ntb)(H_2_ntb)(bipy)]·1.5H_2_O·Guest}_n_ and {[Zn_3_(ntb)_2_(bpa)]·2H_2_O·Guest}_n_ (bpa—1,2-bis(4-pyridyl)diazene) were 3D MOFs with relatively large channels filled with dimethylammonium cations and/or solvent guest molecules. Crystal structure of {(Me_2_NH_2_)_2_[Zn_2_(ntb)_2_(bipy)]·2DMF}_n_ revealed a 2D→3D catenated architecture. Compounds {(Me_2_NH_2_)_2_[Zn_2_(ntb)_2_(py)_2_]·DMF·2H_2_O}_n_, {[Zn_2_(ntb)(H_2_ntb)(bipy)]·1.5H_2_O·Guest}_n_ and {(Me_2_NH_2_)_2_[Zn_2_(ntb)_2_(bipy)]·2DMF}_n_ demonstrated strong ligand-centered emission in the range of 458–524 nm depending on the structure of the co-ligand (Table 6). Surprisingly, coordination polymer {[Zn_3_(ntb)_2_(bpa)]·2H_2_O·Guest}_n_ demonstrated no emission, which was attributed to photoinduced electron transfer from bpa to ntb^3-^ resulting in luminescence quenching [192].

Yang et al. prepared two lanthanide mixed-carboxylate MOFs [Eu(ntb)(ndc)(H_2_O)]_n_ and [Tb(ntb)(ndc)(H_2_O)]_n_ (H_2_ndc—1,4-naphthalenedicarboxylic acid) [193]. The most intensive emission band (614 nm) of Eu-MOF corresponds to the ^5^D_0_→^7^F_2_ transition (λ_ex_ = 402 nm), while for Tb-MOF this band at 544 nm was due to ^5^D_4_→^7^F_5_ transition (λ_ex_ = 374 nm). Somewhat short excited state lifetimes (about 340 μs) for these MOFs were associated to vibronic coupling because of the presence of coordinated water molecules and deprotonated carboxyl groups. Tb-MOF demonstrated a much better quantum yield compared to Eu-MOF (11% against 0.06%), which was attributed to a better match between the triplet energy level of ntb^3-^ and emitting level of Tb^3+^ [193].

Hu et al. reported Cd-MOF {[Cd_2.5_Na(ntb)_2_(dmf)_4_]·3DMF}_n_ with intraligand π*–π emission at 445 nm (λ_ex_ = 370 nm) [194]. Luminescence quenching was observed in the presence of nitrobenzene of 2,4,6-trinitrophenol in MOF suspensions in DMF, the photoinduced electron-transfer was suggested as a possible quenching mechanism.

Similar photophysical behavior was reported for another Cd-ntb MOF [Cd_5_(ntb)_4_(H_2_O)_2_]_n_, for which strong luminescence quenching (λ_em_ = 417 nm, λ_ex_ = 352 nm) was observed in the presence of 2,4,6-trinitrophenol and 2,4-dinitrophenol [195]. It should be noted that the emission response was selective for two mentioned compounds and no quenching was observed in the presence of other nitroaromatic derivatives.

A sensing platform for detection of Al^3+^ and Fe^3+^ ions based on MOF {[Tb_3_(ntb)_2_(dma)_0.5_(OH)_3_(H_2_O)_0.5_]·3H_2_O}_n_ was proposed in [196]. Among 17 tested metal ions, only Fe^3+^ lead to complete quenching of emission at 549 nm (^5^D_4_ → ^7^F_5_ transition) with the detection limit of 8·10^−6^ M in MeOH-H_2_O, 1:4 solution. At the same time, Al^3+^ ion lead to a significant enhancement of emission band at 463 nm, detection limit 7·10^−7^ M [196].

Wen et al. prepared MOF {[Cd_3_(ntb)_2_(bimb)]·6DMA}_n_ (bimb—4,4′-bis(imidazol-1-yl)biphenyl), which demonstrated ligand-centered emission at 470 nm upon excitation at 360 nm [197]. A notable solvatochromism was observed for this MOF and the emission changed from purple (423 nm) in methanol to indigo (461 nm) in water to cyan (490 nm) in ethanol. Since no solvatochromism was detected for the free ligand, the shifts in emission maxima were attributed to the formation of exciplexes between the solvent and MOF excited state.

Solvent-dependent structural diversity of manganese(II) coordination polymers with H_3_ntb was explored in [198]. Four new coordination polymers, [Mn_3_(ntb)(HCOO)_3_(def)_3_]_n_, {[Mn_3_(ntb)_2_(EtOH)_2_]·DMF}_n_, {(Me_2_NH_2_)_2_[Mn_5_(ntb)_4_(H_2_O)_2_]·4DMF·2H_2_O}_n_ and {[Mn_3_(ntb)_2_(py)_4_(H_2_O)]·H_2_O}_n_, were prepared and their photophysical properties were investigated. All of them demonstrated ligand-centered emission in the range of 421–453 nm (Table 6) with coordination-induced shifts relative to the free ligand (λ_em_ = 448 nm, λ_ex_ = 333 nm).

Coordination polymer based on two ligands with extended π-systems, H_3_ntb and 1,1,2,2-tetrakis(4-(pyridin-4-yl)phenyl)ethane (tppe), {[Zn_3_(tppe)_0.5_(ntb)_2_]·4DMA·7H_2_O}_n_ was reported by Wu et al. [199]. Crystal structure of this MOF resembled a 3D framework of {Zn_3_(ntb)_2_} layers interconnected by tppe pillars (Figure 15). MOF demonstrated significant porosity with the opening window sizes of 18.4 × 14.3 Å^2^ and void volume as high as 72.7%. The emission spectrum of this compound comprised a single band at 475 nm, which was assigned to the fluorescence of both ligands in the framework. High porosity of the MOF made it promising for fluorescent detection of nitroaromatic compounds in the vapor phase. Thin films of MOF were fabricated and after exposure to vapors of different aromatic nitro-derivatives showed luminescence quenching, the highest degree of which was achieved in case of mono-nitro compounds, such as 2-nitrotoluene, 2-nitrophenol and nitrobenzene [199].

Xia et al. reported the preparation of Cd-MOF {[Cd_1.5_(ntb)(bpa)_0.5_(EtOH)]·1.5EtOH}_n_ [200]. It demonstrated a remarkable sensitivity to 4-hydroxy-4′-nitrobiphenyl (HNBP), which quenched the emission near 450 nm with the detection limit as low as 50 nM. It should be noted that other biphenyls, including structurally similar to HNBP 4-phenylphenol and 4-nitrobiphenyl caused almost no changes in emission intensity. Molecular force field calculations indicated bonding between HNBP and MOF via close O⋯H contacts and C–H⋯π interactions. In addition, MOF proved to be efficient and recyclable methylene blue photodegradation catalyst [200].

A water-stable indium-organic framework {(Me_2_NH_2_)[In(ntb)_4/3_]·2DMF·3H_2_O}_n_ was reported in [201]. It demonstrated blue emission at 478 nm in the solid state, and after the exchange of dimethylammonium cations to 4-[*p*-(dimethylamino)styryl]-1-ethylpyridinium dye dual emission with maxima at 479 and 590 nm was observed. The obtained dye-encapsulated MOF exhibited sensing towards Hg^2+^ ions, which caused almost complete luminescence quenching with the detection limit of 1.75 ppb. Quenching of luminescence also occurred in the presence of dichromate anions Cr_2_O_7_^2-^ and nitro-explosives in water.

Hu et al. prepared three coordination polymers [Zn(μ_2_-Hntb)(phen)]_n_, {[Cd(μ_3_-Hntb)(phen)]·2H_2_O}_n_, [Mn(μ_2_-Hntb)(phen)(H_2_O)]_n_ (phen—1,10-phenanthroline) [202]. Zn and Cd MOFs showed an intense emission bands at 513 nm and 515 nm attributed to π*→n or π*→π transitions. Luminescence quantum yield of Zn-MOF was higher compared to Cd-MOF, which was explained by heavy atom effect of cadmium.

Photophysical properties of titanium-organic framework ZSTU-1, {[Ti_6_(μ_3_-O)_6_(μ_2_-OH)_6_(ntb)_2_]·H_2_O·2DMF}_n_ were studied by Zhong et al. [203]. Ethanol suspensions of ZSTU-1 demonstrated ligand-centered emission near 460 nm, which was quenched by nitroaromatic compounds, most notably by picric acid (complete quenching at 238 ppm). Luminescence was also quenched by Fe^3+^ ions, the detection limit was determined to be 63.8 nM. A combination of photoinduced electron transfer and fluorescence resonance energy transfer was suggested to be responsible for luminescence quenching.

A pillar-layered MOF {(Me_2_NH_2_)_2_[Cd_3_(ntb)_2_(bdc)]}·4DMF}_n_ was prepared by Wang et al. [204]. MOF exhibited emission at 440 nm in DMF suspension, a red-shift of emission compared to the free ligand DMF solution (424 nm) was attributed to ligand-to-metal charge transfer. The emission was solvent-dependent and was quenched in the presence of nitroaromatic compounds, for which the quenching constants (K_SV_) from Stern–Volmer equation were determined. Picric acid had the highest K_SV_ value of 96.52. Photoinduced electron transfer was proposed as a possible mechanism for fluorescence quenching [204].

An aldehyde-decorated MOF, [Cd_3_(ntb)_2_(apa)]_n_ (apa—2-amino-3-pyridinecarboxaldehyde) was reported in [205]. Post-synthetic oxidation of this MOF by hydrogen peroxide lead to a coordination polymer Cd-TCOOH containing both amino and carboxyl groups simultaneously. Cd-TCOOH in HEPES buffer (pH 7.4) displayed emission at 450 nm. Influence of a wide range of mono-, di- and trivalent metal cations on the luminescence intensity of Cd-TCOOH was evaluated and only for Al^3+^ an emission turn-on effect was detected. The detection limit for Al^3+^ in water of 0.54 ppb is one of the best reported MOF sensors. The detection mechanism was associated with Al^3+^ coordination to Brønsted acidic sites of Cd-TCOOH. Presence of both Brønsted acidic and basic sites in Cd-TCOOH it allowed to selectively detect lysin without interference from other common aminoacids [205].

Zinc(II) and cadmium(II) coordination polymers {[Cd_3_(ntb)_2_(2,2′-bipy)_2_]·3DMF·2EtOH·H_2_O}_n_ and {[Zn_3_(ntb)_2_(2,2′-bipy)_2_(dmf)_2_]·2DMF·2EtOH·H_2_O}_n_ were prepared using 2,2′-bipyridyl (2,2′-bipy) and H_3_ntb as ligands [206]. Their crystal structure resembled 2D sheets cross-linked by π-π interactions of 2,2′-bipy ligands (Figure 16). Intraligand emissions occurred at 532 and 528 nm for Cd and Zn compounds, respectively. Sensing properties of the coordination polymers were explored and it was found that the luminescence intensity somewhat increased in the presence of Na^+^ and Zn^2+^ ions for Cd coordination polymer and Cd^2+^, Na^+^, Zn^2+^ for Zn coordination polymer. Other studied metal ions caused luminescence quenching, the most efficient were Fe^3+^ and Cu^2+^ ions (all measurements were carried out in DMF) [206].

Some derivatives of H_3_ntb, in particular 5-(bis(4-carboxybenzyl)amino)isophthalic acid (H_4_bcbaip, Scheme 7), a four-connected tetrahedral linker, are also used as emissive ligands for the construction of luminescent coordination polymers. Thus, in an anionic indium-organic framework {(Et_2_NH_2_)[In(bcbaip)]·4DEF·4EtOH}_n_ the tetracarboxylate ligand acted as a light-harvesting antenna. Coumarin 343 (C343) and Coumarin 6 (C6) were loaded into MOF cavities and the photoluminescence spectra were measured. Upon excitation at 381 nm the emission of the framework near 450 nm was quenched, while the emission of the dyes (530 nm for C6 and 510 nm for C343) was enhanced, indicating energy transfer from the network to the dyes molecules [207].

Wu et al. reported a 3D framework {[Cd_3_(H_2_O)(bcbaip)_2_][Cd(im)_3_(H_2_O)_2_]·3H_2_O}_n_ with a rare 4,4,8-connected topology [208]. It exhibited an intraligand emission peak at 410 nm upon excitation at 300 nm. The luminescence was completely quenched in the presence of 300 ppm of nitrobenzene or 220 ppm of Fe^3+^ ions in DMF suspensions of MOF.

A series of isostructural lanthanide MOFs {[Ln_2_(bcbaip)_2_(dma)_2_]·nH_2_O}_n_ (Ln = La^3+^, Pr^3+^, Nd^3+^, Sm^3+^, Eu^3+^, Gd^3+^, Tb^3+^ and Tm^3+^) were prepared an characterized by Wang et al. [209]. La^3+^ and Tm^3+^-MOFs demonstrated lanthanide-tuned ligand-centered emission near 420 nm (Table 6), Sm^3+^-MOF exhibited co-luminescence of Sm^3+^ and bcbaip^4−^ ligand at 417 nm. Eu^3+^-MOF showed red emission characteristic for lanthanide-centered excitation. In contrast, Pr^3+^ and Nd^3+^-MOFs demonstrated NIR emission. Eu^3+^-MOF was evaluated for sensing of nitroaromatic compounds in DMA suspensions and the most notable quenching effect was observed for 2,4,6-trinitrophenol. Nd^3+^-MOF demonstrated luminescence quenching response towards benzaldehyde, representing the first example of NIR luminescent sensing of this analyte [209].

Lu et al. reported zinc(II) and cadmium(II) coordination polymers—3D MOFs [Zn_2_(bcbaip)]_n_ and {[Cd_3_(bcbaip)_2_(H_2_O)][Na_2_(μ_2_-H_2_O)(H_2_O)_7_]}_n_, as well as 2D layered {[Zn_2_(bcbaip)(phen)_2_(H_2_O)]·H_2_O}_n_ [210]. MOFs without additional 1,10-phenanthroline ligand exhibited emission near 415 nm (Table 6), while the introduction of 1,10-phenanthroline induced a red-shift of emission to 520 nm. In all cases, the emission bands were assigned to ligand-to-metal charge transfer. Luminescence intensity notably decreased in the presence of nitroaromatic compounds, especially 2,4-dinitrotoluene and 2-nitrotoluene.

Ren et al. reported a bimetallic network [ZnLi_2_(bcbaip)(dmf)(H_2_O)]_n_ (HNU-22) that demonstrated a broad emission from 396 to 480 nm upon excitation at 368 nm [211]. The luminescence was reversibly quenched in the presence of various metal ions, the most pronounced effect was observed for Fe^3+^.

Four new coordination polymers with H_4_bcbaip were prepared by Qiao et al. [212]. Zinc(II), manganese(II) and lead(II) formed 3D MOF [Zn_2_(bcbaip)]_n_, [Mn(H_2_bcbaip)(H_2_O)]_n_ and {[Pb_2_(bcbaip)(H_2_O)]·3H_2_O·DMF}_n_, while in the case of cadmium(II) a 2D polymer {[Cd(H_2_bcbaip)(H_2_O)_2_]·H_2_O}_n_ was obtained. Zn and Pb MOFs demonstrated emission typical for coordination polymers with bcbaip^4−^ linkers with LMCT effect, while the emission spectrum of Cd coordination polymer resembled a three-band profile (Table 6).

A series of lanthanide MOFs were exploded as candidates for the fabrication of warm while light LEDs. Single-lanthanide coordination polymers {(Me_2_NH_2_)_2_[Ln_2_(bcbaip)_2_]·H_2_O}_n_ (Ln = Sm^3+^, Tb^3+^, Dy^3+^) showed emission typical for the corresponding Ln^3+^ ions (Table 6) [213]. Tri-doped Sm_0.1_Tb_0.04_Dy_0.06_-MOF when excited at 389 nm emitted warm while light with CIE coordinates (0.333, 0.3522) and correlated color temperature 4444 K [213].

The above-mentioned series was extended in [214] by the preparation of Eu and Gd MOFs of the same composition. Eu-MOF exhibited the typical ^5^D_0_→^7^F_1_ (591 nm) and ^5^D_0_→^7^F_2_ (614 nm) transitions, while for a mixed-lanthanide Eu_0.02_Dy_0.18_-MOF white emission with CIE coordinates (0.3336, 0.3168) was achieved. In addition, this mixed-metal MOF demonstrated sensing abilities for water in ethanol, on increasing the water concentration band at 614 nm gradually decreased, while the emission intensity of the ligand at 416 nm increased. The ratio of these two intensities linearly correlated with the water content in the range of 0 to 10%, making the MOF perspective for water determination in bio-ethanol fuels [214].

Eu-MOF {(Me_2_NH_2_)_2_[Ln_2_(bcbaip)_2_]·DMA}_n_ demonstrated luminescence quenching in the presence of Fe^3+^ in aqueous suspension with the detection limit of 23 µM [215].

An unusual 3D lithium-organic framework, [Li_4_(bcbaip)(dmf)_2_]_n_ (HNU-31) was prepared and characterized by Feng et al. [216]. It demonstrated emission at 402 nm, which dramatically enhanced in the presence of Al^3+^ ions. Other metal ions did not interfere with the detection and a low detection limit of 4.4 μM was achieved. Since the emission maximum gradually shifted from 402 to 420 nm with the aluminum concentration increase, but no MOF degradation occurred, the authors assumed that Al^3+^ interacts with the framework [216].

Bis(4-carboxybenzyl)amine (H_2_bca, Scheme 7) was used a flexible fluorophoric ligand for the preparation of four coordination polymers {[Zn_2_(bca)_2_(o-bimb)_2_]·(H_2_O)_2_}_n_, {[Pb(bca)(p-bib)_0.5_]·H_2_O}_n_, {[Cd_2_(bca)_2_(p-bib)_2_]·(H_2_O)_3_}_n_ and {[Zn(bca)(m-bib)]·H_2_O}_n_ (p-bib—1,4-bis(imidazol-1-yl)benzene, m-bib—1,3-bis(imidazol-1-yl)benzene, o-bimb—1,2-bis(imidazol-1-ylmethyl)benzene) [217]. Luminescent properties were studied only for Zn compounds and ligand-centered emission bands near 405 nm (Table 6) were detected. Both coordination polymers demonstrated solvent-dependent emission intensity, the highest intensity was observed in methanol and the lowest in acetone suspensions. Pronounced luminescence quenching effect allowed to achieve low acetone detection limit of about 0.1 μM in aqueous suspensions. Chromate CrO_4_^2−^ and dichromate Cr_2_O_7_^2−^ anions had quenching effect with detection limit in the order of 0.1 µM being one of the lowest among MOF sensing materials [217].

The conjugated π-electron system of H_3_ntb can be extended by the introduction of additional phenyl rings. Xu et al. prepared tris(3′-carboxybiphenyl)amine (H_3_tcba, Scheme 7) ligand and lanthanide MOFs {[Ln(tcba)(H_2_O)_2_]_2_·DMF}_n_ (Ln = Tb^3+^, Eu^3+^, Gd^3+^) [218]. Upon excitation at 380 nm Gd-MOF showed emission at 435 nm, which is blue-shifted relative to the free ligand (462 nm). Tb and Eu-MOFs demonstrated characteristic red and green emission, Eu-MOF also showed weak ligand emission near 480 nm (Table 6). Luminescence quenching effect by nitroaromatic compounds was thoroughly studied by DFT and microcalorimetric measurements and it was found that the complementarity of Tb-MOF channels and 2,4,6-trinitrohenol molecules explains their enhanced interaction [218].

Ligand with an even more extended conjugated system, tetraphenylphenylenediamine (H_4_tppd, Scheme 7) was used by Mayer et al. to prepare zinc(II) and cadmium(II) coordination polymers, that demonstrated two-photon-absorption-induced fluorescence (TPPF) [219]. Coordination polymer [Zn_2_(tpbd)(dma)_2_]_n_ resembled a two-dimensional network, while [Cd_2_(tpbd)(H_2_O)_4_]_n_ was a three-dimensional framework. Both coordination polymers demonstrated photoluminescence near 520 nm with relatively high quantum yields of 24% (Zn-CP) and 52% (Cd-CP). The two-photon-absorption cross-section was nine time higher of Cd-CP compared to Zn-CP, which was ascribed to a higher degree of interpenetration in Cd-MOF, leading to higher chromophore density and stronger excitonic interactions.

## 9. Ligands Based on Highly Emissive Ruthenium(II) and Iridium(III) Complexes

Xie and co-workers designed and synthesized highly porous and phosphorescent Ir-containing coordination polymers for oxygen sensing via efficient and reversible luminescence quenching [220]. Carboxylic derivatives of tris(2-phenylpyridine)iridium [Ir(ppy)_3_] were used as linkers to prepare 2D coordination polymers {[Zn_4_(*μ*_4_-O)[Ir(3-cppy)_3_]_2_]·6DMF·H_2_O}_n_, {[Zn_3_[Ir(4-cppy)_3_]_2_(def)_2_(H_2_O)_2_]·DEF·H_2_O}_n_ and {[Zn_3_[Ir(4-cppy)_3_]_2_(dmf)(H_2_O)_3_]·2DMF·3H_2_O}_n_ (3-cppyH_3_—2-(3-carboxyphenyl)pyridine, 4-cppyH_3_—2-(4-carboxyphenyl)pyridine, Scheme 8). All compounds upon excitation at 385 nm (for [Ir(3-cppy)_3_] derivatives) or 400 nm (for [Ir(4-cppy)_3_] derivatives) displayed emission at 538 nm and 565 nm, respectively, corresponding to LMCT transitions. The LMCT phosphorescence can be quenched by molecules with a triplet ground state, thereby providing accurate sensing for oxygen. It was shown that the luminescence of {[Zn_4_(*μ*_4_-O)[Ir(3-cppy)_3_]_2_]·6DMF·H_2_O}_n_ is reversibly quenched by O_2_, in contrast, for coordination polymers based on [Ir(4-cppy)_3_] derivatives phosphorescence quenching was irreversible [220].

Barrett et al. have doped [Ir(ppy)_2_(4,4′-dcppy)], [Ir(ppy)_2_(5,5′-dcbpy)]Cl and [Ru(2,2′-bpy)_2_(5,5′-dcbpy)]Cl_2_ (4,4′-dcppy—6-(4-carboxyphenyl)nicotinic acid, 5,5′-dcbpy—2,2′-bipyridinyl-5,5′-dicarboxylic acid, Scheme 8) complexes into the UiO-67 framework to obtain three highly porous and phosphorescent MOFs {Zr_6_(µ_3_-OH)_4_(µ_3_-OH)_4_(bpdc)_6-x_[Ir(ppy)_2_(5,5′-dcppy)]_x_}_n_ (x = 0.029), {Zr_6_(µ_3_-OH)_4_(µ_3_-OH)_4_(bpdc)_6-x_[Ir(ppy)_2_(5,5′-dcbpy)]_x_}_n_ (x = 0.058) and {Zr_6_(µ_3_-OH)_4_(µ_3_-OH)_4_(bpdc)_6-x_[Ru(2,2′-bpy)_2_(5,5′-dcbpy)]_x_}_n_ (x = 0.179), which exhibited reversible phosphorescence quenching by oxygen [221]. Upon excitation at 420 nm, the emission spectra of MOFs showed emission maxima at 599, 593 and 614 nm, respectively, consistent with the phosphorescent emission from MLCT excited states of the doped ligands.

Representative examples of luminescent Ir–Ln heteronuclear coordination polymers {[Ln[Ir(ppy)_2_(5,5′-dcbpy)]_2_(OH)]·H_2_O}_n_ (Ln = Gd, Yb, Er, Nd) based on a highly efficient light-harvesting Ir antenna were reported in [222]. The Ir unit showed strong visible light absorption via ^3^MLCT and sensitized the Ln(III)-based NIR emission by efficient d→f energy transfer. The NIR emission spectra of the coordination polymers were recorded upon excitation at 500 nm in the solid state. The observed emission band at 1535 nm was attributed to 4I_13/2_→4I_15/2_ transition for the Ir–Er complex, for the Ir–Yb complex the luminescence with a maximum at 980 nm was assigned to 2F_5/2_→2F_7/2_ transition. Three strong emission bands at 880 nm, 1056 nm and 1326 nm, corresponding to the 4F_3/2_→4I_9/2_, 4I_11/2_ and 4I_13/2_ transitions were observed for Ir–Nd complex.

Li et al. synthesized a new iridium-based two-fold-entangled MOF {[Zn[Ir(ppy)_2_(5,5′-dcbpy)]_2_]·4H_2_O}_n_, which demonstrated reversible structural dynamics and luminescence color changes (red to orange) in response to the loss of guest H_2_O molecules [223]. The coordination polymer displayed a strong emission band at 620 nm when excited at 500 nm in the solid state, similar to that of the iridium unit with the maximum emission peak at 628 nm, indicating that the luminescence is generated by the Ir unit arising from ^3^MLCT. The desolvated sample exhibited an orange luminescence at 592 nm, with a distinct 28 nm blue-shift relative to the initial coordination polymer.

Four new lead(II)–iridium(III) heterobimetallic coordination frameworks, {[Pb_2_[Ir(ppy)_2_(4,4′-dcbpy)]_4_(dmf)_2_](ClO_4_)·2DMF·13H_2_O}_n_, {[Pb[Ir(ppy)_2_(4,4′-dcbpy)]_2_(H_2_O)](ClO_4_)·3Me_2_CO·3H_2_O}_n_, {[Pb[Ir(ppy)_2_(4,4′-dcbpy)]_2_(H_2_O)]·3Me_2_CO·3H_2_O·CH_3_CN}_n_ and {[Pb_4_[Ir(ppy)_2_(4,4′-dcbpy)]_4_I_4_(dmf)_2_]·10H_2_O}_n_ (4,4′-dcbpy–2,2′-bipyridine-4,4′-dicarboxylate) were synthesized by Chen et al. [224]. The photophysical characteristics of the coordination polymers are given in Table 7. Lead(II) ions were introduced to the frameworks to promote phosphorescence-based sensitivity to oxygen, all coordination polymers were able to detect oxygen in real gas mixtures.

Three magnesium coordination polymers based on highly luminescent Ir(III) units [Ir(ppy)_2_(4,4′-dcbpy)] were reported in [225]. All coordination polymers, {[Mg[Ir(ppy)_2_(4,4′-dcbpy)]_2_(H_2_O)_2_]·3.5H_2_O}_n_, {[Mg[Ir(ppy)_2_(4,4′-dcbpy)]_2_(dmf)_2_]·3.5H_2_O}_n_ and {[Mg((Ir(ppy)_2_(4,4′-dcbpy))_2_(def)(H_2_O)]·3H_2_O}_n_ represented allomeric 1D chain structures. All compounds demonstrated blue shifts of luminescence maxima (Table 7) compared to the free Ir(III) unit, which displayed a strong orange luminescence near 575 nm upon excitation at 468 nm.

The first examples of phosphorescent coordination polymers formed through the self-assembly between the iridium complexes *rac*-, Λ- and Δ-[Ir(mesppy)_2_(qpy)]PF_6_ (mesppy—2-phenyl-4-mesitylpyridine, qpy—4,4′:2′,2′′:4′′,4′′′-quaterpyridine) and Ag^+^ ions through qpy–Ag coordination were reported by Martir et al. [226]. Enantiopure Λ- and Δ-IrAg formed porous nanospheres resembling the shape of marigold flowers. The emission bands of the coordination polymers *rac*-, Λ- and Δ-IrAg at 646 nm were slightly broader and red-shifted, with slightly higher quantum yields of 15%, 14% and 15% and slightly longer lifetimes of 375, 382 and 379 ns compared to the starting metalloligands.

Three iridium(III)-based MOFs, namely {[Cd_3_[Ir(3-cppy)_3_]_2_(dmf)_2_(H_2_O)_4_]·6H_2_O·2DMF}_n_, {[Cd_3_[Ir(3-cppy)_3_]_2_(dma)_2_(H_2_O)_2_]·0.5H_2_O·2DMA}_n_ and {[Cd_3_[Ir(3-cppy)_3_]_2_(def)_2_(H_2_O)_2_]·8H_2_O·2DEF}_n_, were synthesized by Fan and coworkers [227]. All coordination polymers were isostructural 3D frameworks. The solid-state photoluminescence emission spectrum of {[Cd_3_[Ir(3-cppy)_3_]_2_(dmf)_2_(H_2_O)_4_]·6H_2_O·2DMF}_n_, (λ_ex_ 380 nm) showed a single peak at 519 nm in contrast to two maxima at 488 and 510 nm for Ir unit, corresponding to the ^3^MLCT transition. The phosphorescence lifetime of this MOF (2.95 μs) was longer than that of the parent linker (1.93 μs). MOF exhibited excellent water stability which made it a highly selective and sensitive multiresponsive luminescent sensor for Fe^3+^ and Cr_2_O_7_^2−^. The detection limits were 67.8 and 145.1 ppb, respectively. This MOF could also be used as an optical sensor for selective sensing of adenosine triphosphate (ATP^2−^) over adenosine diphosphate (ADP^2−^) and adenosine monophosphate (AMP^2−^) in an aqueous solution.

A phosphorescent MOF {[Zn_2_(bpdc)_2_[Ru(2,2′-bpy)_2_(5,5′-dcbpy)]_0.5_]·9DMF·9H_2_O}_n_ contains [Ru(2,2′-bpy)_2_(5,5′-dcbpy)]^2+^ as a linker was reported in [228]. Diffusion-controlled luminescence quenching was studied using a series of different amines as quenchers. MOF was excited at 452 nm, and the emission intensity at 627 nm was recorded at different time points after the addition of a predetermined amount of amine quenchers. Triethylemine (TEA), tripropylamine (TPA) and tribulylamine (TBA) can diffuse through the MOF channels according to the time-dependent quenching data, whereas diisopropylethylamine is too large to enter the MOF channels. Diffusivities of TEA, TPA and TBA in MOF could were determined to be (1.1 ± 0.2)·10^−13^, (4.8 ± 1.2)·10^−14^ and (4.0 ± 0.4)·10^−14^ m^2^/s, respectively.

Liu and coworkers synthesized phosphorescent nanoscale coordination polymers using the [Ru(2,2′-bpy)_2_(5,5′-dcbpy)] bridging ligand and Zn^2+^ or Zr^4+^ nodes [229]. The UV/vis spectrum of Zn coordination polymer in ethanol showed a broad MLCT absorption band between 400 and 550 nm. It exhibited an emission maximum at 635 nm with a luminescence quantum yield of 2.1% and an average lifetime of 215 ns. Zirconium coordination polymer showed emission at 630 nm with a luminescent quantum yield of 0.8% and an average luminescence lifetime of 107 ns. In vitro viability assays for Zr nanoscale coordination polymer on H460 human non small-cell lung cancer cells were conducted and a significant MLCT luminescent signal was observed in the confocal z section images for H460 cells.

Zhang et al. synthesized a highly symmetric microporous MOF {(Me_2_NH_2_)[In[Ru(4,4′-dcbpy)_3_]]·6H_2_O}_n_, which manifested a broad visible light MLCT absorption band between 250 and 650 nm [230]. The free ligand displayed a strong red photoluminescence with the emission maximum at 633 nm upon excitation at 400 nm, and MOF emission was centered at 657 nm. The red shift was attributed to the coordination of ligands with In(III) centers. The fluorescence lifetime of the ligand was determined to be 512 ns, while it is found to be 301 ns for the coordination polymer. Photocatalytic activity of MOF in visible light-induced photodegradation of methyl orange was demonstrated. Moreover, sensing properties of CP were also evaluated, and the result shows that CP can selectively detect the nitro explosives molecules.

Porous luminescent complexes {[Mg_2_[Ru(4,4′-dcbpy)_3_]]·13H_2_O}_n_, {[Mg_2_[Ru(5,5′-dcbpy)_3_]]·19H_2_O}_n_, {[Sr_4_[Ru(4,4′-dcbpy)_3_]_2_]·18H_2_O}_n_, {[Sr_2_[Ru(5,5′-dcbpy)_3_]]·14H_2_O}_n_ and {[Cd_2_[Ru(5,5′-dcbpy)_3_]]·12H_2_O}_n_ were prepared by Kobayashi and coworkers [231]. Ruthenium ligands showed absorption bands at 467 and 484 nm and emission bands at 633 nm (for 4,4′-dcbpy derivatives) and 668 nm (for 5,5′-dcbpy derivatives), due to the singlet and triplet MLCT. All of the obtained coordination polymers showed a very broad absorption band below 600 nm and an emission band at ~680 nm (Table 7). The emission lifetimes were in the same time scale as those of [Ru] ligands.

Zhang et al. successfully prepared a fascinating 3D hierarchical flower-like nanostructure of a functional Ru–polypyridine incorporated MOF {[Cd_2_[Ru(4,4′-dcbpy)_3_]]·12H_2_O}_n_ [232]. Absorption spectra showed that MOF exhibited efficient visible light harvesting with the absorption edge extended to around 650 nm due to the MLCT. The luminescence lifetime of MOF was 5.49 µs, which is more than one order of magnitude greater than that of [Ru] unit (483 ns). MOF could promote the visible light induced photocatalytic reduction of CO_2_ to HCOO^−^. The quantum yield of HCOO^−^ reaches 0.67% under 475 nm irradiation.

Two Ru−polypyridine based MOFs, {(Me_2_NH_2_)_2_[Cd_3_[Ru(5,5′-dcbpy)_3_]_2_}_n_ and {[Cd[Ru(4,4′-dcbpy)_2_(2,2′-bpy)]]·3H_2_O}_n_ with non-interpenetrated and 2-fold interpenetrated structures were reported in [233]. The solid-state absorption spectra of MOFs demonstrated very broad absorption bands between 400 and 650 nm due to the ^1^MLCT of [Ru] metalloligands. MOF displayed long luminescence decay lifetimes of 4.98 μs and 6.45 μs, respectively. The interpenetrated MOF exhibited remarkable thermal stability and superior photostability for photocatalytic CO_2_ reduction as compared to its non-interpenetrated counterpart. Both MOFs exhibited similar photocatalytic activities for HCOO^−^ generation, showing the HCOO^−^ production of 16.1 μmol and 17.2 μmol in 6 h, respectively. The quantum yield of HCOO^−^ was about 0.5% under 475 nm irradiation.

Watanabe and coworkers synthesized a new porous coordination polymer {[La_1.75_(OH)_1.25_[Ru(4,4′-dcbpy)_3_]]·16H_2_O}_n_ [234]. The coordination polymer exhibited a dark-red broad emission centered at 691 nm without any vibronic progressions, which is largely shifted to the longer wavelength than that of the ^3^MLCT. A reversible structural transition triggered by water adsorption/desorption was observed—the emission band shifted to the longer wavelength (by about 14 nm) after drying at 110 °C, the wavelength of the emission maximum gradually shifted to the shorter wavelength with increasing humidity. At 100% humidity, the luminescence was restored.

Xu et al. demonstrated the first example for generating an electroluminescent signal from a redox-active MOF {[Zn[Ru(4,4′-dcbpy)_2_(2,2′-bpy)]]·2DMF·4H_2_O}_n_ [235]. MOF was immobilized on the graphene oxide (GO) electrode surface. The maximum emission of the ligand [Ru(4,4′-H_2_dcbpy)_2_(2,2′-bpy)] and MOF were observed at 689 nm and 671 nm. The authors demonstrated the efficiency of the MOF-GO system for the determination of cocaine in serum sample.

Polapally et al. synthesized a new MOF, {[Ru(4,4′-H_2_dcbpy)Cu(4,4′-dcbpy)(4,4′-Hdcbpy)_2_(H_2_O)]·5H_2_O}_n_ [236]. The UV–vis diffuse reflectance spectra of MOF showed the characteristic broad band with two peaks at 440 nm and 480 nm due to the MLCT. These bands were red-shifted compared to the discrete [Ru(4,4′-H_2_dcbpy)_3_]^2+^, which showed a broad band centered at 427 nm. The emission of MOF in the solid state was near 605 nm when excited at 440 nm, which is shifted to shorter wavelength compared to ligand (622 nm). Copper(II) centers showed ferromagnetic interactions with θ = 34 K and C = 0.43 emu·mol^−1^·K for its Curie–Weiss equation [236].

Two new porous coordination polymers {[Ln_7_(OH)_5_[Ru(4,4′-dcbpy)_3_]_4_]·4H_2_O}_n_ (Ln = Ce^3+^, Nd^3+^) were reported in [237]. Cerium(III) coordination polymer exhibited a broad dark red emission with the maximum at 684 nm and vapochromic effect associated with water vapor adsorption/desorption, arising from the ^3^MLCT emission of Ru ligand. Upon excitation at 420 nm, Nd(III) coordination polymer showed emission at 884, 1054 and 1336 nm assignable to the 4f−4f emission of the Nd^3+^ center (^4^F_3/2_→^4^I_9/2_, ^4^I_11/2_ and ^4^I_13/2_, respectively). In the excitation spectra (λem = 1054 nm), the ^1^MLCT absorption bands in the visible region were clearly observed in addition to the sharp 4f−4f transitions.

Kobayashi et al. synthesized proton-conductive and phosphorescent porous coordination polymers, based on Ru(bpy)_3_-type ligands functionalized with six phosphonate groups, {[La_3_[H_5.5_Ru(dpbpy)_3_]_2_]·6H_2_O}_n_ and {[Pr_3_[H_5.5_(Ru(dpbpy)_3_]_2_]·4.5H_2_O}_n_ (H_4_dpbpy—2,2′-bipyridine-4,4′-bis(phosphonic acid) were reported in [238]. It was shown that the porous structures collapsed on the removal of water molecules from the channels, but the original porous structure was reconstructed upon water adsorption. Both MOFs exhibited emission bands near 640 nm similar to that of Ru ligand due to the ^3^MLCT of the Ru ligand. The emission band underwent a blue-shift on increasing the relative humidity from 0 to 75% and from 0 to 54%, respectively. The emission spectra obtained at 100% humidity were almost identical to those of the as-synthesized samples.

## 10. Porphyrin Derivatives

Porphyrins and their derivatives play an important role in chemistry and biology; they participate in light-harvesting, oxygen transport and catalytic transformations [239,240,241,242]. Ligand based on porphyrins benefit from their rigid scaffold, photophysical properties and allow to obtain a wide range of coordination polymers with diverse luminescent properties [243,244,245].

### 10.1. Tetrakis(4-Carboxyphenyl)porphyrin

Tetrakis(4-carboxyphenyl)porphyrin (H_6_tcpp, Table 8) is a D_4h_-symmetric tetratopic linker, which is widely used to construct MOFs. Rigid square geometry and free rotation of the terminal phenyl ring make it possible to obtain MOFs with various topologies. Zr-based MOF porphyrins have several topologies since the connectivity of Zr_6_ clusters can be tuned to 6-, 8- and 12-connetcted topologies, e.g., PCN-224 (**she**), PCN-222/ MOF-545 (**csq**) and MOF-525 (**ftw**) [246].

PCN-222 or MOF-545 formulated as [Zr_6_(μ_3_-OH)_8_(OH)_8_(H_2_tcpp)_2_]_n_ has a 3D network based on Zr−O clusters connected by tcpp^4-^ linkers. PCN-222 shows two emission maxima—a strong peak at 652 nm and a weaker peak at 718 nm when excited at 420 nm (Table 8). These data are consistent with the emission of the free H_6_tcpp monomer. PCN-222(Zr) can be used as a biosensor for label-free detecting phosphoproteins (*casein a*) [247], PCN-222-Pd(II) worked as a turn-on sensor for detecting Cu(II) ions [248].

Deibert and Li demonstrated a pH-dependent “turn-off-turn-on” effect for PCN-222 in acidic solutions. The turn-off effect was found for peaks in the range from 645 to 760 nm at HCl concentration of more than 0.01 M. At pH value of 0, a turn-on effect was observed for the peak at 525 nm, it was significantly enhanced in combination with the disappearance of the shoulder peak at 465 nm, and the peak at 490 nm shifted to 500 nm [249].

Ye et al. used PCN-222(Zn) for detecting pentachlorophenol. The addition of pentachlorophenol to a suspension of PCN-222(Zn) quenches the luminescence at 629 nm when excited is 449 nm. The sensor demonstrated high selectivity and sensitivity, fast and linear luminescent response, the detection limit was determined to be as low as 33 ppb [250].

Morris et al. reported the synthesis and crystal structure of MOF-525, a zirconium(IV) framework [Zr_6_O_4_(OH)_4_(H_2_tcpp)_3_]_n_ with **ftw** topology [251]. The emission maximum of MOF-525 was observed at 651 nm upon excitation at 512 nm, which can be assigned to the typical S_1_→S_0_ transition of the porphyrin scaffold. Li et al. demonstrated the use of MOF-525 as a turn-off sensor for detecting of Cu^2+^ ions with the detection limit reaching 67 nM [252].

Deria et al. showed the dependence between the luminescent properties of the framework and its topology [268]. They compared NU-902, formulated as [Zr_6_(μ_3_-O)_4_(μ_3_-OH)_4_(OH)_4_(H_2_O)_4_(H_2_tcpp)_2_]_n_ (rhombic, **scu)** and MOF-525 (cubic, **ftw**) as well as their metallated derivatives NU-902(Zn) and MOF-525(Zn). According to the DFT optimization, MOF-525 contains non-planar porphyrin due to the non-coplanar arrangement of the *meso*-phenyl groups with respect to the carboxylate groups and the porphyrin macrocycle, and the interchromophoric center-to-center distance is 13.5 Å. In contrast, the linker in NU-902 adopts a stable conformation, and the carboxylates and *meso*-phenyl are almost coplanar, the interchromophoric distance is 10.5 Å. The emission maxima of MOF-525 and NU-902 have different intensities and are red-shifted relative to H_6_tcpp. The luminescence spectra of MOF-525(Zn) and NU-902(Zn) were more similar and featured a slight red-shift compared to H_4_tcpp(Zn). The authors explained it by the interchromophoric interaction, which depends on the topology of the framework. A stronger interchromophoric interaction leads to a smaller S_1_–S_0_ energy gap with faster emissive state radiative decay and a larger red shift [268].

Feng et al. reported a series of highly stable PCN-224(M) MOFs, formulated as [Zr_6_(OH)_8_(Mtcpp)]_n_ (M = 2H, Co, Ni, Fe) [269]. PCN-224 has a (4,6)-connected **she** topology with large 3D channels. Upon excitation at 590 nm, a suspension of PCN-224 exhibited a strong emission peak at 651 nm with a shoulder at 705 nm, which is due to the S_1_→S_0_ transition of porphyrin. PCN–224 was used as a turn-off sensor for detecting 2,4,6-trinitrotoluene [253] or Hg^2+^ ions [270] in aqueous solutions with fast response, high sensitivity and selectivity.

Two more Zr-based porphyrinic MOFs, [Zr_6_(μ_3_-O)_4_(μ_3_-OH)_4_(OH)_4_(H_2_O)_4_(H_2_tcpp)_2_]_n_ or PCN-225 and [Zr_6_(μ_3_-O)_4_(μ_3_-OH)_4_(OH)_4_(H_2_O)_4_(Zntcpp)_2_] or PCN-225(Zn) were reported in [254]. Both featured two types of open channels and represented (4.8)-connected **sqc** networks. The excitation of PCN-225 at 415 nm lead to the emission near 725 nm, which was assigned to the tccp^4−^ fluorescence. The fluorescence intensity of PCN-225 correlated with the pH values. In acidic solutions, low-intensity fluorescence was observed, while the most intense fluorescence was detected at pH 10.2. The fluorescent response was most sensitive in the pH range from 7 to 10, which was explained by protonation-deprotonation of pyrrole rings in the porphyrin ligand [254].

Chen and Fukuzumi [255] reported a 2D coordination polymer [Zn(Me_4_tcpp)]_n_, in which each porphyrinic unit is linked to four other via Zn-O bonds. Upon excitation at 558 nm, it demonstrated two emission bands at 604 and 648 nm, similar to a discrete coordination compound [Zn(Et_4_tcpp)]_n_.

The dependence between the photoluminescent properties and the topology of MOFs was discussed in [256]. Five different MOFs based on H_6_tcpp or H_4_Mtcpp, {[Pb_2_(H_2_tcpp)]∙4DMF∙H_2_O}_n_, {Pb_2_(Cotcpp)(H_2_O)(dmf)]∙1.5DMF}_n_, {[Pb_2_(Nitcpp)(dmf)(H_2_O)]∙1.5DMF∙2H_2_O}_n_, {[Pb_2_(Cutcpp)(dmf)(H_2_O)]∙1.5DMF∙2H_2_O}_n_, {[Pb_2_(VOtcpp)(H_2_O)_2_]∙4DMF}_n_ were synthesized. All compounds had different framework topologies and the conformation of porphyrin rings varied from flat to wavy to bowl-shaped. The conformation of the porphyrin cores affected on their stacking, the organization of the tcpp ligands and the coordination spheres of Pb^2+^ cations. {[Pb_2_(tcpp)]_4_∙4DMF∙H_2_O}_n_ exhibited three emission bands at 403, 436 and 467 nm, while other MOFs showed two emission maxima near 400 nm and 450 nm, when excited at 290 nm [256].

Hou et al. obtained isomorphic 3D MOFs [Sr_2_(H_2_tcpp)(dmf)_4_]_n_ and [Ba_2_(H_2_tcpp)(dmf)_4_]_n_ [257]. Upon excitation at 420 nm, both MOFs demonstrated strong solid-state emission with a peak at 662 nm and a shoulder at 711 nm, which are blue-shifted compared to H_6_tcpp. The fluorescence intensity and the position of the emission maximum for [Ba_2_(H_2_tcpp)(dmf)_4_]_n_ depended on the nature of the solvent and its dielectric constant.

Fateeva et al. reported a water stable MOF {[Al_2_(OH)_2_(H_2_tcpp)]·3DMF·2H_2_O}_n_ [258]. The fluorescence spectrum of this MOF coincided with the fluorescence spectrum of the free linker. The emission maximum at 660 nm was due to S_1_→S_0_ transition of the porphyrin. Metalation of the porphyrin by Zn^2+^ ions lead to a change in the fluorescence spectra and the appearance of another maximum at 620 nm.

Coordination polymer [Zr(tbaPy)_5_(tcpp)]_n_ (tbaPy–1,3,6,8-tetra(4-carboxylphenyl)pyrene, Table 8) was reported in [271]. This MOF exhibited low fluorescence intensity, however, it could be used as a turn-on sensor for detection H_2_S or S^2-^.

Spoerke et al. used MOF [Zn_2_(Zntcpp)(4,4′-bpy)_1.5_]_n_, PPF-4 for the fabrication of dye-sensitized solar cells, in which isolated MOF crystals were used as the sensitizer. The deconvolution of the photoluminescence spectrum of PPF-4 showed that it consists of three peaks at 618, 633 and 664 nm, that are blue-sifted compared to Zn-tcpp (λ_ex_ 590 nm) [259].

### 10.2. Other Porphyrin Derivatives with the Carboxylic Groups

Johnson et al. reported four isostructural MOFs, UNLPF-10a, -10b, -11 and -12 based on 3,5-bis[(4-methoxycarbonyl)phenyl]phenylporphine (Table 8) and its derivatives metallated by In^3+^, [SnCl_2_]^2+^ and Sn^4+^ [260]. All MOFs exhibited two emission maxima (Table 8). UNLPF-12 demonstrated high photostability and photocatalytic activity in the reactions of aerobic hydroxylation of arylboronic acids, amine coupling and Mannich reaction.

TBP-MOF based on tetrakis(4-carboxyphenyl)tetrabenzoporphyrin (H_6_tbp, Table 8) and {Zr_6_O_4_(OH)_4_(C_6_H_5_COO)_2_} clusters was reported in [261]. TBP-MOF exhibited fluorescence peaks at 705 and 780 nm, which were red-shifted by about 100 nm compared to PCN-224. TBP-MOF could be used as an effective photodynamic therapy agent.

Two MOFs based on 5,15-di(p-benzoato)porphyrin (H_2_dbp, Table 8) and its partially reduced derivative 5,15-di(p-benzoato)chlorin (H_2_dbc), [Hf_6_(μ_3_-O)_4_(μ_3_-OH)_4_(dbp)_6_]_n_ (DBP−UiO) [272] and [Hf_6_(μ_3_-O)_4_(μ_3_-OH)_4_(dbc)_6_]_n_ (DBC−UiO) [262] were reported by Lu and co-authors. DBP-UiO showed negligible fluorescence compared to H_2_dbp, which showed two peaks at 630 nm (strong) and 690 nm (weak) (λ_ex_ = 405 nm). H_2_dbc exhibited emission at 641 nm of high intensity, while the fluorescence of DBC-UiO was weak. Both MOFs could be used for highly efficient photodynamic therapy of resistant head, neck or colon cancer.

Demel and et al. synthesized a series of MOFs, {[Ln*_2_*(OH)_4.7_(Por)_0.33_]∙2H_2_O}_n_ (Ln = Eu^3+^, Tb^3+^; Por = 5,10,15,20-*tetrakis*(4-sulfonatophenyl)porphyrin (H_2_tpps^4-^) or Pdtpps^4-^) [263], their emission maxima are shown in Table 8.

### 10.3. Porphyrins Functionalized with Nitrogen-Containing Heterocycles

Porphyrins functionalized with nitrogen-containing heterocycles can also be used to construct porphyrinic frameworks.

ZnMPyTPP is the first example of MOF based on (5-pyridyl-10,15,20-triphenylphorphyrinato)zinc(II) (ZnTPP, Table 8). It featured a 1D zig-zag chain structure [264]. The emission maxima of ZnMPyTPP were detected at 603 and 649 nm (λ_ex_ = 410 nm), which is similar to the data for ZnTPP. An increase in concentration leads to a red-shift of the band at 603 nm by 10 nm, which was explained by polymerization.

Pan et al. prepared 2D MOF [Fe(tpyp)]_n_ with alternating sequences ABAB and ABCDABCD based on *meso*-tetra(4-pyridyl)porphyrin (tpyp, Table 8). The luminescence spectrum of [Fe(tpyp)]_n_ contained two bands at 668 and 716 nm (λ_ex_ = 410 nm) [265].

Son et al. reported coordination polymers [Zn_2_(tcpb)(DAZnP)]_n_, DA-MOF and [Zn_2_(tcpb)(FZnP)]_n_, F-MOF (H_4_tcpb—1,2,4,5-tetrakis(4-carboxyphenyl)benzene, DAZnP—[5,15-bis[(4-pyridyl)ethynyl]-10,20-diphenylporphinato]zinc(II), F-ZnP—[5,15-di(4-pyridyl)-10,20-bis(pentafluorophenyl)porphinato]zinc(II), Table 8) [266]. Upon excitation at 400 nm, the emission peak of F-MOF was detected at 680 nm, while DA-MOF had no detectable emission. Later, the same group obtained quantum dots based on these two coordination polymers with improved light harvesting properties [273].

A series of coordination polymers based on 5,10,15,20-tetrakis(4,4′-dipyridylaminophenylene)porphyrin (H_2_tdpap) were reported in [274]. [Mn^III^Mn^II^(tdpap)Cl_3_(dmf)]_n_ had a 2D coordination network, the layers of which were linked by π···π stacking interactions into a 3D structure. {[Cu_4_(tdpap)(AcO)_5_(HCOO)(AcOH)(H_2_O)_3_]·AcOH·H_2_O}_n_ featured 2D sheets composed of 50- and 70-membered metallomacrocycles. [Zn_3_(tdpap)(AcO)_4_]_n_ had 1D zigzag coordination chains linked to form a 2D structure. {[Cd_2_(H_2_tdpap)(AcO)_4_]·DMF·AcOH·2H_2_O}_n_ had a stair-like 2D structure formed by the linkage of 1D coordination chains through the bridging of binuclear [Cd_2_(AcO)_4_]_n_ subunits. The photoluminescent properties of coordination polymers were dependent on their structure and metal coordination. The free ligand demonstrated two emission peaks at 679 nm and 726 nm upon excitation at 480 nm. Compounds containing Mn^2+^, Mn^3+^ and Cu^2+^ did not have detectable emission due to the paramagnetic effects. Cd coordination polymer demonstrated a weaker emission compared to Zn compound due to the heavy atom effect. A blue-shifted peak at 668 nm for Zn coordination polymer is explained by violation of the aromaticity of porphyrin due to the asymmetric coordination of Zn^2+^ [274].

Liu et al. synthesized MOF PCN-526 based on 5,10,15,20-tetrakis[4-(2H-tetrazol-5-yl)phenyl]porphyrin (Table 8). PCN-526 demonstrated emission at 407 nm upon excitation at 323 nm. The introduction of large fluorescent molecules (anthracene, phenanthrene, perylene, etc.) into PCN-526 pores affected the energy transfer pathways, which lead to changes in the luminescent properties [267].

## 11. Conclusions

To summarize, currently, a wide variety of highly emissive ligands is available to researchers, enabling them to tune the dimensionality and topology of the coordination polymers, as well as their functional properties, including sensing ability, thermochromism and photochromism, electroluminescence. Among the highly-emissive ligands, di- and polycarboxylate linkers (4,4′-stilbenedicarboxylic acid, bis(3,5-dicarboxyphenyl) derivatives of naphthalene diimide, 4,4′,4′′-nitrilotribenzoic acid), are available, allowing the assembly of coordination polymers based on most metal ions (3d transition metals, lanthanides, Mg^2+^, Ca^2+^, Sr^2+^). N,N-donor ligands with pronounced luminescent properties, such as bis(4-pyridyl)naphthalene diimide, 4,7-bis(4-pyridyl)-2,1,3-benzothiadiazole, terpyridine derivatives, are also accessible for the construction of frameworks with metal ions capable of forming strong bonds with nitrogen donor atoms (Zn^2+^, Cd^2+^, Cu^+^, Ag^+^). In addition, they can serve as auxiliary ligands in combination with polycarboxylate linkers to form pillar-layered metal-organic frameworks.

Despite the large number of luminescence data published for the coordination polymers, most works limit the results to the excitation and emission maxima, only some works report quantum yields, even in fewer works time-resolved photoluminescence experiments were carried out. The emission mechanisms in majority of the works are proposed tentatively, so the luminescence mechanisms of the coordination polymers are yet to be studied in detail using modern experimental techniques and DFT calculations.

It should be noted that derivatives of BODIPY dyes, ruthenium(II) and iridium(III) complexes, which demonstrate strong luminescence in the free state, but surprisingly, the coordination polymers based on these linkers did not exhibit high quantum yields (10% and less).

Many works that describe the synthesis of luminescent coordination polymers consider their sensory properties. However, it should be noted that the range of analytes studied is not very wide - in most cases these are simple aromatic compounds (benzene, toluene) and their nitro derivatives (nitrobenzene, di- and trinitrotoluene, nitrophenols), or a limited set of inorganic ions (Fe^3+^, Al^3+^, Cr_2_O_7_^2−^). Sensing mechanisms are studied only in minor part of the works, so detailed evaluation of reasons for luminescent response, as well as inclusion of new analytes (biologically relevant molecules, environmental pollutants) seem to be promising areas for further research.

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
