# Peer review of "Coordination Polymers Based on Highly Emissive Ligands: Synthesis and Functional Properties"

_materials, 2020, doi:10.3390/ma13122699_

Round 1
Reviewer 1 Report
Potapov et al. have conceived an attracting collection of coordination polymers incorporating fluorescent ligands from the classes of 4,4’-stilbenedicarboxylic acid, 1,3,4-oxadiazole, thiazole, 2,1,3-benzothiadiazole, terpyridine and carbazole derivatives, naphthalene diimides, 4,4’,4’’-nitrilotribenzoic acid, ruthenium(II) and iridium(III) complexes, BODIPY derivatives, and porphyrins. The manuscript, although too exhaustive, can thus be accepted for publication after considering some of the following minor points:
- line 28 (from Keywords): instead of 'naphthalene diimide; 4,4’-stilbenedicarboxylic acid; ligands; carbazole; thiazole' the authors should write 'emissive ligands'.
- line 53: instead of 'nitric oxide (II)' should be 'nitric(II) oxide', while carbon monoxide does not need the specification of carbon valence (II).
- line 71: since nothing related to the synthesis of coordination polymers discussed in the review, I suggest removal of this word.
- In the tables depicting many types of ligands, the authors should add a column to show the emission wavelength of free ligands, just a clearer comparison with the corresponding coordination polymers.
Author Response
1. line 28 (from Keywords): instead of 'naphthalene diimide; 4,4’-stilbenedicarboxylic acid; ligands; carbazole; thiazole' the authors should write 'emissive ligands'.
We believe that the keywords that we have chosen can improve the visibility of the paper for the indexing systems, however, in the revised version we have substituted the keyword “ligands” to a more relevant term “emissive ligands”, as suggested by the reviewer.
2. line 53: instead of 'nitric oxide (II)' should be 'nitric(II) oxide', while carbon monoxide does not need the specification of carbon valence (II).
Corrected
3. line 71: since nothing related to the synthesis of coordination polymers discussed in the review, I suggest removal of this word.
The sentence was rephrased and now does not mention the synthesis.
4. In the tables depicting many types of ligands, the authors should add a column to show the emission wavelength of free ligands, just a clearer comparison with the corresponding coordination polymers.
Data for the free ligands were added to additional rows in tables 1, 5, 6 and to separate columns in tables 2, 3, 4, 7, 8.
Reviewer 2 Report
Manuscript is generally fairly well written database-like paper on known MOF structures with the focus being placed on structures of ligands, although Authors seem to keep an eye on properties (mostly fundamental ones, like fluoresence / luminescence). Indeed, Authors constructured their review taking particular structures of ligands as a criterion, which in my opinion is the easiest approach possible, and also - rather old-fashioned.This is simply because it tends to discriminate ligands that are not so frequently used for MOF synthesis, but yield spectacular results. On the bright side, it should be noted that Authors did not limit themselves to well known ligands like 4,4'-stilbenedicarboxylic acids, but also directed attention to emergent groups of ligands, like BODIPY. I appreciate the work done by Authors.
Nevertheless, I have certain reservations regarding the coverage of content provided in this review paper, hence I recommend major revision of its content.
- Honestly speaking, the field of "classic" (by classic I mean the frames of linear optics, that is employing one-photon excitation) emission properties of CPs was reviewed many, many times. In the meantime nonlinear optical properties of coordination polymers started to gain attention, as new experimental techniques appeared, allowing to characterize them in facile manner. Thus, currently there is more than a handful of examples of new CPs, whose luminescence is excited via two-photon or even three-photon absorption, the latter of which was used e.g. for NIR-to-VIS luminescent remote temperature sensing. See papers of Vittal, Fischer, Samoć. There is already quite a lot of papers on NLO-based luminescence of CPs and MOFs, but authors mentioned only one example by Quah at et.
- Conclusions section is extremely short and lacks any deeper idea! Review paper should end with some interesting ideas, relationships, maybe even generalizations. Authors should try deriving possible directions and challenges that are expected for the field - in other words something more original than they already wrote in conclusions section.
Author Response
- Honestly speaking, the field of "classic" (by classic I mean the frames of linear optics, that is employing one-photon excitation) emission properties of CPs was reviewed many, many times. In the meantime nonlinear optical properties of coordination polymers started to gain attention, as new experimental techniques appeared, allowing to characterize them in facile manner. Thus, currently there is more than a handful of examples of new CPs, whose luminescence is excited via two-photon or even three-photon absorption, the latter of which was used e.g. for NIR-to-VIS luminescent remote temperature sensing. See papers of Vittal, Fischer, Samoć. There is already quite a lot of papers on NLO-based luminescence of CPs and MOFs, but authors mentioned only one example by Quah at et.
We agree that nonlinear optical properties of the coordination polymers is a rapidly developing field of research and deserves a separate review, two of which have actually been already published in 2017 and 2018 (refs. 44 and 47 in the manuscript). We have included several recent works from the groups mentioned by the Reviewer in our manuscript as references 45, 46, 48, 49, 154, 219.
- Conclusions section is extremely short and lacks any deeper idea! Review paper should end with some interesting ideas, relationships, maybe even generalizations. Authors should try deriving possible directions and challenges that are expected for the field - in other words something more original than they already wrote in conclusions section.
The section was substantially extended (lines 1469-1493) and now includes generalizations on the types of ligands used in combination with certain methods, as well as comments on the possible directions for advancing the studies of luminescent coordination polymers.
Reviewer 3 Report
Authors have presented a report on existing publications on Luminescent coordination polymers from the angle of emissive ligands. The review has quite elaborately and extensively covered the literature. However, in current form, it does not provide a clear overview/reasoning from the perspective of scientific mechanisms involved with these emissive ligands. I have few suggestions for the authors to improve it to an acceptable form from a reader's point of view.
1) Since, the review articles generally aim to provide a good overview of the field and should be readable to people with different level of expertise. Authors should provide fundamental overview of luminescence and MOFs/Coordination polymers with explanation/directions of mechanisms which play a role in these materials in introduction. Of-course this should not too much in detail that the text digress from the focus of the area. But it can bring readers to complete understanding of the concepts.
2) Authors have classified the materials from the point of different organic linkers with emissive properties. I see good observation of results throughout from literature. However, the manuscript in current state keeps the discussions superficially. It could have been expanded to explanation of properties with respect to scientific mechanisms like MLCT, LMCT, LLCT, their effects on HUMO-LUMO levels, reasons for turn-on, turn-off effects etc. Based on these, suggestions could be made to develop the design strategies based on these mechanisms in different ligand groups which affect the structural and functional properties.
3) I also find that it misses a comparison of these emissive ligand groups even though these materials have been individually described in the manuscript. Thus, for a researcher, while tuning the properties of the MOFs with respect to functional groups on ligands, I could not see that the manuscript covers the existing knowledge for design of luminiscent MOFs with clear suggestions/reasons for choosing a particular ligand group. If that was not the purpose, then I clearly miss the reason for the classification with respect to ligand groups.
4) Moreover, it would be beneficial if the tables, describing different materials with linkers and associated photo-physical properties, also indicate the associated mechanism for the emission for clear overview in tables.
In current form, I see that the manuscript requires extensive changes to incorporate the suggestions and thus, I can not suggest the acceptance of the manuscript in current form.
Author Response
1) Since, the review articles generally aim to provide a good overview of the field and should be readable to people with different level of expertise. Authors should provide fundamental overview of luminescence and MOFs/Coordination polymers with explanation/directions of mechanisms which play a role in these materials in introduction. Of-course this should not too much in detail that the text digress from the focus of the area. But it can bring readers to complete understanding of the concepts.
We have added an introduction into the concept of luminescence in general and in relevance to the coordination polymers (lines 44-65) and Figure 1.
2) Authors have classified the materials from the point of different organic linkers with emissive properties. I see good observation of results throughout from literature. However, the manuscript in current state keeps the discussions superficially. It could have been expanded to explanation of properties with respect to scientific mechanisms like MLCT, LMCT, LLCT, their effects on HUMO-LUMO levels, reasons for turn-on, turn-off effects etc. Based on these, suggestions could be made to develop the design strategies based on these mechanisms in different ligand groups which affect the structural and functional properties.
It should be noted that the luminescence mechanisms are not discussed in a substantial part of works and in most works can be considered speculative. Nevertheless, in addition to those already mentioned in the original manuscript we have added the mechanisms for emission and/or sensing properties of the coordination polymers for all works in which they were discussed (lines 117, 156, 197, 313, 357, 367, 372, 380, 386, 416, 451, 466, 474, 477, 504, 512, 522, 534, 542, 555, 564, 566, 568, 607, 615, 620, 635, 637, 643, 648, 655, 673, 679, 696, 713, 803, 819, 846, 858, 881, 886, 891, 899, 917, 922, 975, 977, 986, 990, 1019, 1036, 1038, 1106).
3) I also find that it misses a comparison of these emissive ligand groups even though these materials have been individually described in the manuscript. Thus, for a researcher, while tuning the properties of the MOFs with respect to functional groups on ligands, I could not see that the manuscript covers the existing knowledge for design of luminiscent MOFs with clear suggestions/reasons for choosing a particular ligand group. If that was not the purpose, then I clearly miss the reason for the classification with respect to ligand groups.
We have added the discussion on ligand types in the conclusions.
4) Moreover, it would be beneficial if the tables, describing different materials with linkers and associated photo-physical properties, also indicate the associated mechanism for the emission for clear overview in tables.
Since not all works report the mechanisms, we decided not to overcrowd the tables, but add the mechanisms in the manuscript text.
Round 2
Reviewer 2 Report
I am impressed by the work done by Authors in such a short timeframe. As far as I can see, they responded to all my comments, as well as they appear to address comments of other Referees. Conclusions section is now as broad as it deserves to be for this type of rather an extensive work.
Taken together, I am pretty sure that this paper will gain attention from the CP community, which should be also reflected in citations.
Reviewer 3 Report
Authors have addressed the comments of the reviewers pretty well in such a short period of time. I see that both introduction and conclusion sections are expanded in a constructive and improvised manner. The other changes in the manuscript are also acceptable. I agree with the remaining comments to my remarks. It can be accepted in current manner. However, small grammatical check is required which authors should review once during proof-check.
Congratulations for such extensive work and I believe that this will be a good addition to the literature.